# Dual Privacy Protection in Decentralized Learning

## Abstract

In decentralized learning systems, significant effort has been devoted to protecting the privacy of each agent's local data or gradients. However, the shared model parameters themselves can also reveal sensitive information about the targets, which the network is estimating. While differential privacy-based decentralized learning can protect network estimates, using excessively large privacy noise variance will significantly reduce the accuracy of network estimates. To this end, we propose a dual-protection framework for decentralized learning. Within this framework, we develop two privacy-preserving algorithms, named DSG-RMS and EDSG-RMS. Different from existing differential privacy distributed learning methods, the designed algorithms simultaneously obscure the network's estimated values and local gradients, by adding a protective perturbation vector at each update and by using random matrix-step-sizes. Then, we establish convergence guarantees for both algorithms under convex objectives. In particular, our error bound and privacy analysis highlight how the variance of the random matrix-step-sizes affects both algorithmic performance and the privacy of local gradients. Despite using large-variance random step-sizes for stronger gradient privacy, the network's estimation accuracy in our algorithms can still be improved by choosing a sufficiently small algorithmic parameter $\gamma$. Finally, we validate the practical effectiveness of the proposed algorithms through extensive experiments across diverse applications, including distributed filtering, distributed learning, and target localization.

## 1 Introduction

In decentralized learning, a key task is estimating global parameters from local data across distributed agents, as in cooperative spectrum sensing, multi-target localization, and bio-inspired systems Sayed (2022). Agents collaborate via incremental, consensus, or diffusion strategies, exchanging intermediate results or gradients. Such exchanges, however, can compromise privacy, since gradients may reveal sensitive local training data Shokri & Shmatikov (2015); Ma et al. (2023).

To address privacy concerns in decentralized learning, various protective mechanisms have been developed. Cryptographic approaches—such as secret sharing Li et al. (2019a), secure multi-party computation Mohassel & Zhang (2017), and homomorphic encryption Lu & Zhu (2018); Ruan et al. (2019); Fu et al. (2024)—offer strong privacy guarantees, but incur substantial computational and communication overhead. System decomposition methods enhance privacy by constructing virtual agent and restructuring local objective; however, they introduce additional computational complexity Zhang et al. (2018). Differential privacy (DP) provides a lightweight alternative by injecting zero-mean noise into shared quantities He et al. (2018); Wei et al. (2020); Hu et al. (2024). It has been successfully integrated into ADMM-based distributed algorithms and gradient tracking frameworks across both directed and undirected network topologies Zhang & Zhu (2016); Huang et al. (2024); Zhu et al. (2018); Lü et al. (2020). Another class of noise-based methods applies multiplicative noise to modify local measurements, as seen in Harrane et al. (2016); however, its effectiveness relies on the restrictive assumption that local optima coincide with the global solution, and it significantly increases communication overhead. The fundamental challenge of DP-based decentralized learning algorithms lies in the inherent trade-off between privacy and network accuracy: stronger privacy requires more noise, which in turn degrades the quality of network estimated parameters. Recent work has sought to improve this trade-off through techniques such as variance-decaying noise, zero-sum noise, and graph-homomorphic noise, while achieving $(\epsilon, \delta)$-differential privacy Ding et al.

(2021); Rizk et al. (2023). However, in distributed consensus learning settings, it faces a critical phenomenon: As the iteration proceeds, the transmitted signals become nearly identical—for example, in decentralized stochastic gradient algorithm. When the added noise has a small variance—but sufficient to preserve gradient—an adversary may still recover the mean value of transmitted data by applying statistical operations such as sliding-window averaging. This privacy risk is particularly acute in wireless sensor localization networks, where the shared data typically contains sensitive location information Piperigkos et al. (2021); Shi et al. (2022).

In this paper, we focus on the privacy risks related to the exposure of network estimated values and local gradients/data during information exchange. To address these concerns, we introduce a novel dual-protection privacy-enhancing framework that integrates two key components: a non-zero protection vector and a random matrix-step-sizes (RMS) mechanism. By embedding this framework into the decentralized stochastic gradient (DSG) algorithm and exact diffusion variant (EDSG), we develop two advanced privacy-preserving methods: DSG-RMS and EDSG-RMS. Then, we conduct a comprehensive convergence analysis of the proposed algorithms under convex objective functions. The theoretical results demonstrate that both DSG-RMS and EDSG-RMS achieve convergence to a neighborhood of the optimal solution. Furthermore, we examine the effect of random matrix-step-sizes and protection vectors on algorithmic performance. Notably, increasing the variance of the random step-sizes enhances gradient privacy but simultaneously amplifies sensitivity to data heterogeneity, resulting in looser error bounds and degraded network estimation accuracy. Our main contributions are summarized as follows:

- We propose two novel algorithms for decentralized learning: DSG-RMS and EDSG-RMS. The EDSG-RMS variant is particularly well-suited for settings with heterogeneous data across devices.

- We provide rigorous convergence analyses for both algorithms under convex and strongly convex objective functions. This analysis reveals how specific parameter choices influence the overall network performance. Even when a large variance in the random matrix-step-sizes is used to ensure strong privacy of the gradient, the proposed algorithms can improve the network's estimation accuracy by reducing the parameter $\gamma$.

- We conduct comprehensive experiments, and the results confirm the effectiveness of our methods. We further evaluate their ability to preserve privacy, showing that both network estimates and individual data remain well protected.

## 2 RELATED WORKS

As discussed earlier, the PD offers a way to protect shared gradient information by adding random noise. When this noise typically has a zero mean and and insufficient variance, an adversary can recover the underlying model parameters using simple techniques such as sliding-window averaging. A recent method using masked diffusion attempts to address this issue, but it also employs a zero-mean noise Han et al. (2025). When more zero-mean noise is injected, a small forgetting factor becomes necessary to mitigate its adverse effects, which inevitably weakens the collaborative performance among nodes over network. Our approach takes a different direction. Instead of relying on zero-mean noise, we introduce nonzero perturbation vectors into transmission values and, more generally, employ random matrix-step-sizes for local gradients. This approach enhances privacy protection while preserving learning performance. In addition, our EDSG-RMS algorithm reduces communication overhead—each iteration requires only half as many communication rounds as the masked diffusion primal-dual stochastic gradient algorithm.

In Li et al. (2019b), the NIDS algorithm employs heterogeneous step-sizes that vary only across agents; however, each agent uses a fixed step-size over time. In contrast, our proposed algorithms introduce random matrix-valued step-sizes that vary both across agents (in space) and across iterations (in time). Consequently, every agent utilizes a distinct step-size at each iteration, ensuring that even if the average step-size used by an agent is disclosed, the actual gradient information remains protected. While the work in Wan et al. (2023) also leverages random step-sizes for gradient protection, it operates within a federated learning framework—a centralized setting—and does not provide convergence analysis. Our work focuses on a fully decentralized setting (in both homogeneous and heterogeneous networks) and includes rigorous convergence guarantees for both convex and strongly convex objectives.

## 3 BACKGROUND AND MOTIVATION

Consider the following distributed optimization problem over an undirected network:

$$\min_{w \in \mathbb{R}^L} \ J(w) = \frac{1}{K} \sum_{k=1}^{K} J_k(w), \tag{1}$$

where $K$ is the number of networked agents, $J_k(w)$ is the convex local risk function at agent $k$.

### 3.1 DSG AND EDSG ALGORITHMS

To solve the problem in a distributed manner, we consider the following two algorithms.

#### 3.1.1 DSG ALGORITHM

At agent $k$, the DSG is executed as

$$\begin{cases} \boldsymbol{\psi}_k(n) = \boldsymbol{w}_k(n-1) - \gamma \widehat{\nabla J}_k(\boldsymbol{w}_k(n-1); \boldsymbol{x}_k(n)), & \text{(local update)} \tag{2a} \\ \boldsymbol{w}_k(n) = \sum_{\ell \in \mathcal{N}_k} a_{\ell k} \boldsymbol{\psi}_\ell(n), & \text{(combination)} \tag{2b} \end{cases}$$

for $n \geq 1$, where the initial weight $\boldsymbol{w}_k(0)$ can be any finite value, $\gamma > 0$ is a deterministic step-size, $A = [a_{\ell k}]$ is a symmetric and doubly stochastic combination matrix, $\widehat{\nabla J}_k(\boldsymbol{w}_k(n-1); \boldsymbol{x}_k(n))$ represents the stochastic gradient using sample $\boldsymbol{x}_k(n)$, and $\mathcal{N}_k$ denotes the set of neighboring agents for agent $k$, including itself.

#### 3.1.2 EDSG ALGORITHM

At agent $k$, the update of EDSG is

$$\begin{cases} \boldsymbol{\psi}'_k(n) = \boldsymbol{w}'_k(n-1) - \gamma \widehat{\nabla J}_k(\boldsymbol{w}'_k(n-1); \boldsymbol{x}_k(n)), & \text{(local update)} \tag{3a} \\ \boldsymbol{\phi}'_k(n) = \boldsymbol{\psi}'_k(n) + \boldsymbol{w}'_k(n-1) - \boldsymbol{\psi}'_k(n-1), & \text{(correction)} \tag{3b} \\ \boldsymbol{w}'_k(n) = \sum_{\ell \in \mathcal{N}_k} a_{\ell k} \boldsymbol{\phi}'_\ell(n), & \text{(combination)} \tag{3c} \end{cases}$$

for $n \geq 1$, where $A = [a_{\ell k}]$ is another symmetric and doubly stochastic combination matrix[1]. The initial values are set as $\boldsymbol{\psi}'_k(0) = \boldsymbol{w}'_k(0)$, where $\boldsymbol{w}'_k(0)$ can take any finite value.

The DSG and EDSG algorithms belong to the well-established classes of primal and primal-dual methods, respectively. Compared to gradient tracking algorithms, both exhibit lower communication overhead, as each iteration entails only a single round of variable exchange among agents. This communication efficiency motivates our focus on these two algorithms. Moreover, as demonstrated in Sayed (2022), EDSG generally outperforms DSG in heterogeneous data settings, whereas DSG tends to be more effective when the data across agents is homogeneous.

### 3.2 PRIVACY ISSUES DISCUSSION

We now examine the privacy issues of these two algorithms under two types of adversaries: external eavesdroppers and honest-but-curious internal agents. External eavesdroppers are passive adversaries located outside the network who intercept communication links to infer agents' estimated parameters, local gradients, and potentially their private training data. Honest-but-curious agents faithfully execute the algorithm but may misuse the information received during collaboration to infer sensitive data—such as gradients or training samples—belonging to their neighbors. In both cases, we assume that both types of adversaries have full knowledge of the combination matrix used in the consensus step.

---

[1]The matrix $A$ in (3) is positive-definite, as detailed in Sayed (2022).

### 3.2.1 PRIVACY RISKS ASSOCIATED WITH $w_k(n)$ AND $w'_k(n)$

If the sequences $\{\boldsymbol{\psi}_\ell(n)\}$ or $\{\boldsymbol{\phi}'_\ell(n)\}$ is accessible to external eavesdroppers for $n = 1, 2, \ldots, N$ and $\ell = 1, 2, \ldots, K$, then the values $\{\boldsymbol{w}_k(n), n = 1, 2, \ldots, N\}$ and $\{\boldsymbol{w}'_k(n), n = 1, 2, \ldots, N\}$ can be derived using (2b) and (3c). This exposure could lead to the unintended disclosure of sensitive network estimation, for instance, the leakage of location information in distributed localization systems.

### 3.2.2 PRIVACY RISKS IN LOCAL GRADIENT INFORMATION

If honest-but-curious agents have access to $\{\boldsymbol{\psi}_\ell(n)\}$ or $\{\boldsymbol{\phi}'_\ell(n)\}$ for $n = 1, 2, \ldots, N$ and $\ell = 1, 2, \ldots, K$, they can infer local gradient information using the relationships $\gamma \widehat{\nabla J}_\ell(\boldsymbol{w}_\ell(n); \boldsymbol{x}_\ell(n+1)) = \boldsymbol{w}_\ell(n) - \boldsymbol{\psi}_\ell(n+1)$ and $\gamma \widehat{\nabla J}_\ell(\boldsymbol{w}'_\ell(n); \boldsymbol{x}_\ell(n+1)) = \boldsymbol{w}'_\ell(n) - \boldsymbol{\psi}'_\ell(n+1)$ for $n = 2, 3, \ldots, N-1$. Then, honest-but-curious agents, knowing the step-size parameter $\gamma$, can accurately reconstruct gradient information from neighboring agents. External eavesdroppers, however, would obtain gradients with an unknown amplitude scaling.

In distributed least-mean-square (LMS) filtering Sayed (2014), the local gradient at time $n$ is expressed as $\gamma(\boldsymbol{d}_k(n) - \boldsymbol{x}_k^\top(n)w)\boldsymbol{x}_k(n)$, where $\boldsymbol{x}_k(n)$ represents the local data. This gradient is a scaled version of the local data, which means that external eavesdroppers may extract sensitive information about the underlying data.

### 3.2.3 PRIVACY RISKS IN LOCAL DATA EXPOSURE

In deep learning, a honest-but-curious agent who has access to both the gradient $\widehat{\nabla J}_k(\boldsymbol{w}_k(n); \boldsymbol{x}_k(n+1))$ and model parameter $\boldsymbol{w}_k(n)$ can leverage model inversion inference techniques, such as those proposed in Zhu et al. (2019), to reconstruct the local training data. Consequently, in the standard DSG and EDSG algorithms, local training data may be inadvertently leaked.

## 4 PROPOSED METHODS

To address the privacy concerns discussed, this section introduces the DSG-RMS and EDSG-RMS algorithms. We then present their mean-square stability and privacy analysis, followed by a discussion of an efficient approach for selecting matrix step-sizes with reduced computational complexity.

### 4.1 PROPOSED ALGORITHMS

We begin by defining a random matrix $\boldsymbol{M}_k(n)$, constructed as follows:

$$\boldsymbol{M}_k(n) = \begin{bmatrix} \boldsymbol{\mu}_{k1}(n) & & & \\ & \boldsymbol{\mu}_{k2}(n) & & \boldsymbol{Z}_k(n) \\ & & \ddots & \\ \boldsymbol{Z}'_k(n) & & & \boldsymbol{\mu}_{kL}(n) \end{bmatrix}, \ L \times L \tag{4}$$

where the blocks $\boldsymbol{Z}_k(n)$ and $\boldsymbol{Z}'_k(n)$ consist of elements with zero mean, while the main diagonal elements $\{\boldsymbol{\mu}_{k\ell}(n), \ell = 1, 2, ..., L\}$ share a common mean value $\mu > 0$. Each element in the random matrix may have a different variance.

Using this random matrix, we develop the following algorithms.

- (DSG-RMS)

$$\begin{cases} \boldsymbol{\psi}_k(n) = \boldsymbol{w}_k(n-1) - \gamma \boldsymbol{M}_k(n) \widehat{\nabla J}_k(\boldsymbol{w}_k(n-1); \boldsymbol{x}_k(n)), & \text{(local update)} & (5a) \\[2mm] \boldsymbol{\psi}_k^c(n) = \boldsymbol{\psi}_k(n) + \dfrac{\tau}{\sqrt{L}} \boldsymbol{c}_k(n-1), & \text{(protection)} & (5b) \\[2mm] \boldsymbol{w}_k(n) = \displaystyle\sum_{\ell \in \mathcal{N}_k} a_{\ell k} \boldsymbol{\psi}_\ell^c(n) - \boldsymbol{\psi}_k^c(n) + \boldsymbol{\psi}_k(n), & \text{(combination)} & (5c) \end{cases}$$

where $\boldsymbol{c}_k(n-1) = \|\boldsymbol{w}_k(n-1)\| \cdot \mathbb{1}_L$ and $\tau \neq 0$ is a free parameter.

- (EDSG-RMS)

$$
\begin{cases}
\boldsymbol{\psi}'_k(n) = \boldsymbol{w}'_k(n-1) - \gamma \boldsymbol{M}_k(n) \widehat{\nabla J}_k(\boldsymbol{w}'_k(n-1); \boldsymbol{x}_k(n)), & \text{(local update)} & \text{(6a)} \\
\boldsymbol{\phi}'_k(n) = \boldsymbol{\psi}'_k(n) + \boldsymbol{w}'_k(n-1) - \boldsymbol{\psi}'_k(n-1), & \text{(correction)} & \text{(6b)} \\
\boldsymbol{\phi}'^c_k(n) = \boldsymbol{\phi}'_k(n) + \dfrac{\tau}{\sqrt{L}} \boldsymbol{c}'_k(n-2), & \text{(protection)} & \text{(6c)} \\
\boldsymbol{w}'_k(n) = \displaystyle\sum_{\ell \in \mathcal{N}_k} a_{\ell k} \boldsymbol{\phi}'^c_\ell(n) - \boldsymbol{\phi}'^c_k(n) + \boldsymbol{\phi}'_k(n), & \text{(combination)} & \text{(6d)}
\end{cases}
$$

where $\boldsymbol{c}'_k(n) = \|\boldsymbol{w}'_k(n)\| \cdot \mathbb{1}_L$ and the initial value $\boldsymbol{c}'_k(-1)$ is set to a random vector.

In (5b) and (6c), $\boldsymbol{c}_k(n-1)$ and $\boldsymbol{c}'_k(n-2)$ serve as data protection vectors, which protect transmission values $\boldsymbol{\psi}_k(n)$ and $\boldsymbol{\phi}'_k(n)$, respectively. Increasing the parameter $\tau$ strengthens this protection but may affect the stability of the algorithm. Theorems 1 and 2 specify the effective range of $\tau$. At each time step $n$, $\gamma \boldsymbol{M}_k(n)$ works as random matrix-step-sizes, with its expected value given by $\gamma \mu \boldsymbol{I}_L$. In the EDSG-RMS algorithm, we use $\boldsymbol{c}'_k(n-2)$ instead of $\boldsymbol{c}'_k(n-1)$, primarily to facilitate the theoretical analysis of the algorithm.

**Remark 1** (*Protection mechanisms*) *During the local update phase, the random matrix $\boldsymbol{M}_k(n)$ serves as a form of multiplicative noise that modifies the gradient information. This matrix is locally generated and remains private to each agent, thereby helping to obscure the true stochastic gradient and reduce the risk of model inversion attacks. In the protection step, a dynamic non-zero vector is added to the transmission vector to prevent inference attacks based on statistical analysis, such as estimating the mean of network updates. Beyond the protection vector form in (5b), alternative formulations can be employed, such as $\tau \mathrm{erf}(0.1\boldsymbol{w}_k(n-1))$ and $\tau \tanh(\boldsymbol{w}_k(n-1))$, where $\mathrm{erf}(\cdot)$ and $\tanh(\cdot)$ denote the error and hyperbolic tangent functions, respectively.*[2]

**Remark 2** (*Method extension*) *The multiplicative noise $\boldsymbol{M}_k(n)$ in (5a), protection mechanism (5b), and combination step (5c) can be integrated into gradient tracking type algorithms to ensure privacy protection. However, unlike the algorithms (5) and (6), which require only one communication round per iteration, the gradient tracking algorithm necessitates two rounds per iteration.*

### 4.2 CONVERGENCE AND PRIVACY ANALYSIS

**Assumption 1** (*Network topology*) *The undirected network is strongly-connected. If agents $\ell$ and $k$ are linked, then $a_{\ell k} > 0$; otherwise $a_{\ell k} = 0$, where $A$ is a symmetric and doubly stochastic matrix.*

*Strong connectedness means that, for any pair of agents, there exists an undirected path with positive edge weights connecting them, and moreover, at least one agent has a positive self-loop (i.e., $a_{kk} > 0$, for some $k$). The combination matrix $\mathcal{A} = A \otimes \boldsymbol{I}_L$ can be decomposed as follows:Sayed (2022)*

$$
\mathcal{A} = \underbrace{[K\Gamma, \quad c\mathcal{X}_R]}_{\mathcal{X}} \underbrace{\begin{bmatrix} \boldsymbol{I}_L & \boldsymbol{0} \\ \boldsymbol{0} & \mathcal{D} \end{bmatrix}}_{\Sigma} \underbrace{\begin{bmatrix} \Gamma^{\mathsf{T}} \\ \frac{1}{c}\mathcal{X}_L \end{bmatrix}}_{\mathcal{X}^{-1}}, \tag{7}
$$

*where $\Gamma = \frac{1}{K}\mathbb{1}_K \otimes \boldsymbol{I}_L$, $\mathcal{D} = \mathsf{diag}\{\lambda_2, \ldots, \lambda_K\} \otimes \boldsymbol{I}_L$, and $c > 0$. The eigenvalues $\{\lambda_2, \ldots, \lambda_K\}$ exclude 1. For DSG-RMS, $-1 < \lambda_\ell < 1$; for EDSG-RMS, $0 < \lambda_\ell < 1$ ($\ell = 2, \ldots, K$). Here, $\boldsymbol{I}_L$ and $\mathbb{1}_K$ denotes $L \times L$ identity matrix and $K \times 1$ all one vector, respectively.*

**Assumption 2** (*Gradient noise*) *For any agent $k$ and time $n$, the gradient noise $\boldsymbol{s}_{k,n}(\boldsymbol{w}) = \widehat{\nabla J}_k(\boldsymbol{w}; \boldsymbol{x}_k(n)) - \nabla J_k(\boldsymbol{w})$ is temporally and spatially independent, and satisfies*

$$
\mathbb{E}\{\boldsymbol{s}_{k,n}(\boldsymbol{w})|\boldsymbol{\mathcal{F}}_{n-1}\} = \boldsymbol{0}, \quad \mathbb{E}\{\|\boldsymbol{s}_{k,n}(\boldsymbol{w})\|^2 |\boldsymbol{\mathcal{F}}_{n-1}\} \leq \sigma^2_{s,k}, \tag{8}
$$

*where $\boldsymbol{w} \in \boldsymbol{\mathcal{F}}_{n-1}$, $\sigma^2_{s,k} \geq 0$, and $\boldsymbol{\mathcal{F}}_{n-1} = \mathrm{filtration}\{\boldsymbol{w}_k(0), \cdots, \boldsymbol{w}_k(n-1), \text{all } k\}$.*

---

[2]The mean-square error analysis for these alternatives can be conducted using inequalities $|\mathrm{erf}(0.1x) - \mathrm{erf}(0.1y)\|^2 \leq 0.1|x - y|^2$ and $|\tanh(x) - \tanh(y)|^2 \leq |x - y|^2$.

**Assumption 3** (*Random matrix-step-sizes*) *For each agent $k$ and time $n$, $\boldsymbol{M}_k(n)$ has mutually independent entries, which are also independent across both time and agents. Its $\ell$-th diagonal element $\boldsymbol{\mu}_{k,\ell}(n)$ satisfies*

$$\mathbb{E}\{\boldsymbol{\mu}_{k,\ell}(n)\} \triangleq \mu, \quad \mathbb{E}\{(\boldsymbol{\mu}_{k,\ell}(n) - \mu)^2\} \triangleq \sigma_{\mu,k}^2, \tag{9}$$

*with constants $\mu > 0$, $\sigma_{\mu,k} > 0$. Off-diagonal entries are zero-mean with variances bounded by $\sigma_z^2$.*

*It follows that $\mathbb{E}\{\boldsymbol{M}_k(n)\} = \mu\boldsymbol{I}_L$, $\mathbb{E}\{\|\boldsymbol{M}_k(n) - \mu\boldsymbol{I}_L\|^2\} \leq \sigma_\mu^2$, and $\mathbb{E}\{\|\boldsymbol{M}_k(n)\|^2\} \leq \theta_\mu^2$, where $\sigma_\mu^2 = \max\{\sigma_{\mu,k}^2 + (L-1)\sigma_z^2, k = 1, 2, ..., K\}$ and $\theta_\mu^2 = \max\{\mu^2 + \sigma_{\mu,k}^2 + (L-1)\sigma_z^2, k = 1, 2, ..., K\}$. Unlike Zhao & Sayed (2014); Han et al. (2025), the variables $\{\boldsymbol{\mu}_{k,\ell}(n)\}$ may take negative values.*

**Assumption 4** (*Lipschitz continuous gradient*) *Each risk function $J_k(w)$ is $\delta$-smooth:*

$$\|\nabla J_k(x) - \nabla J_k(y)\| \leq \delta\|x - y\|, \quad \forall x, y \in \mathbb{R}^L \tag{10}$$

*for some positive constant $\delta$. Additionally, the network cost function $J(w) = \frac{1}{K}\sum_{k=1}^{K} J_k(w)$ is lower bounded, i.e., $J(w) \geq J^*$, where $J^*$ denotes the optimal value of $J(w)$.*

**Assumption 5** (*Convex function*) *Each risk function $J_k(w)$ is convex, meaning that for any $x, y \in \mathbb{R}^L$, the following inequality holds:*

$$J_k(x) - J_k(y) + \frac{\nu}{2}\|x - y\|^2 \leq \langle \nabla J_k(x), x - y \rangle, \tag{11}$$

*where $\nu \geq 0$ is a constant. Let $w^*$ denote an optimal solution. If $\nu > 0$ (i.e., the function is strongly-convex), the optimal solution $w^*$ will be unique.*

**Theorem 1** (**Convergence of DSG-RMS**) Under Assumptions 1–5, the following convergence guarantees hold.

- For the convex case ($\nu = 0$), if $\gamma$ and $\tau$ satisfy

$$\gamma \leq \frac{\mu(1 - \|\mathcal{D}\|)}{14\delta\theta_\mu^2}, \tag{12}$$

$$|\tau| < \frac{1 - \|\mathcal{D}\|}{\sqrt{8}\|\mathcal{D} - \boldsymbol{I}_{(K-1)L}\|}, \tag{13}$$

then the time-averaged expected cost function satisfies

$$\frac{1}{N}\sum_{n=1}^{N}\left(\mathbb{E}\{J(\overline{\boldsymbol{w}}_{n-1})\} - J(w^*)\right) \leq \frac{2\mathbb{E}\{\|\overline{\boldsymbol{w}}_0 - w^*\|^2\}}{\gamma\mu N} + \frac{12\delta\mathbb{E}\{\|\mathcal{W}_0\|^2\}}{N(1 - \|\mathcal{D}\|)K}$$

$$+ \frac{12\gamma^2\theta_\mu^2\delta\|\mathcal{D}\|^2}{1 - \|\mathcal{D}\|}\left(\frac{8\|\nabla\mathcal{J}(\mathcal{W}^*)\|^2}{(1 - \|\mathcal{D}\|)K} + \sigma_s^2\right) + \frac{\gamma}{\mu K}\left(\frac{8\sigma_\mu^2\|\nabla\mathcal{J}(\mathcal{W}^*)\|^2}{K} + \frac{\theta_\mu^2\sigma_s^2}{2}\right), \tag{14}$$

where $\overline{\boldsymbol{w}}_n = \Gamma^{\mathsf{T}}\mathcal{W}_n$, $\mathcal{W}_n = \text{col}\{\boldsymbol{w}_1(n), \boldsymbol{w}_2(n), ..., \boldsymbol{w}_K(n)\}$, $\mathcal{W}^* = \mathbb{1}_K \otimes w^*$, and $\sigma_s^2 = \max\{\sigma_{s,k}^2, k = 1, 2, ..., K\}$.

- For the strongly-convex case ($\nu > 0$), if $\tau$ satisfies the condition (13) and $\gamma$ satisfies

$$\gamma \leq \frac{\mu\nu(1 - \|\mathcal{D}\|)}{64\theta_\mu^2\delta^2}\sqrt{\frac{\nu}{\delta}}, \tag{15}$$

then the expected squared error is bounded as follows:

$$\mathbb{E}\{\|\overline{\boldsymbol{w}}_n - w^*\|^2\} \leq \left(1 - \frac{\gamma\mu\nu}{8}\right)^n\left(\mathbb{E}\{\|\overline{\boldsymbol{w}}_0 - w^*\|^2\} + \frac{\mathbb{E}\{\|\mathcal{W}_0\|^2\}}{K}\right)$$

$$+ \frac{8\gamma}{\mu\nu K}\left(\frac{4\sigma_\mu^2\|\nabla\mathcal{J}(\mathcal{W}^*)\|^2}{K} + \theta_\mu^2\sigma_s^2\right) + \gamma^2\frac{48\theta_\mu^2\delta\|\mathcal{D}\|^2}{\nu(1 - \|\mathcal{D}\|)}\left(\frac{8\|\nabla\mathcal{J}(\mathcal{W}^*)\|^2}{(1 - \|\mathcal{D}\|)K} + \sigma_s^2\right). \tag{16}$$

*Proof:* The proof is provided in the Appendix.

**Theorem 2** (**Convergence of EDSG-RMS**) Under Assumptions 1–5, the following results hold.

- For the convex case ($\nu = 0$), if $\gamma$ and $\tau$ satisfy

$$\gamma \leq \min\left\{ \frac{(1 - \|\mathcal{D}\|)\sigma_b^{0.5}}{28\delta(\sigma_\mu + \mu)}, \frac{\mu}{2\delta(2\mu^2 + \sigma_\mu^2)} \right\}, \tag{17}$$

$$|\tau| \leq \frac{(1 - \|\mathcal{D}\|)\sigma_b^{0.5}}{\sqrt{32(1 - \sigma_b)}}, \tag{18}$$

where $\sigma_b = \min\{\lambda_i, i = 2, 3, ..., K\}$, then the following convergence bound holds

$$\frac{1}{N}\sum_{n=1}^{N}\left(\mathbb{E}\{J(\overline{\boldsymbol{w}}'_{n-1})\} - J(w^*)\right) \leq \frac{2\mathbb{E}\{\|\overline{\boldsymbol{w}}'_0 - w^*\|^2\}}{\gamma\mu N} + \frac{48\delta(3 - \|\mathcal{D}\|)\mathbb{E}\{\|\mathcal{W}'_0\|^2\}}{NK(1 - \|\mathcal{D}\|)^2}$$

$$+ \frac{96\gamma^2\mu^2\delta}{NK}\frac{\mathbb{E}\{\|\nabla\mathcal{J}(\overline{\boldsymbol{\mathcal{W}}}'_0)\|^2\}}{(1 - \|\mathcal{D}\|)^2} + \frac{144\gamma^2\delta\|\mathcal{D}\|^2}{(1 - \|\mathcal{D}\|)}\left(\frac{3\sigma_\mu^2\|\nabla\mathcal{J}(\mathcal{W}^*)\|^2}{(1 - \|\mathcal{D}\|)K} + \theta_\mu^2\sigma_s^2\right)$$

$$+ \frac{2\gamma}{\mu K}\left(\frac{4\sigma_\mu^2}{K}\|\nabla\mathcal{J}(\mathcal{W}^*)\|^2 + \theta_\mu^2\sigma_s^2\right), \tag{19}$$

where $\overline{\boldsymbol{w}}'_n = \Gamma^{\mathsf{T}}\boldsymbol{\mathcal{W}}'_n$, $\boldsymbol{\mathcal{W}}'_n = \text{col}\{\boldsymbol{w}'_1(n), \boldsymbol{w}'_2(n), ..., \boldsymbol{w}'_K(n)\}$, and $\overline{\boldsymbol{\mathcal{W}}}'_n = (\frac{1}{K}\mathbb{1}_K\mathbb{1}_K^{\mathsf{T}} \otimes \boldsymbol{I}_L)\boldsymbol{\mathcal{W}}'_n$.

- For the strongly-convex case ($\nu > 0$), if $\tau$ satisfies the condition (18) and $\gamma$ satisfies

$$\gamma \leq \min\left\{ \frac{(1 - \|\mathcal{D}\|)\sigma_b^{0.5}}{40(\sigma_\mu + \mu)\delta}\sqrt{\frac{\nu}{\delta}}, \frac{\mu\nu(1 - \|\mathcal{D}\|)}{64(\sigma_\mu^2 + \mu^2)\delta^2} \right\}, \tag{20}$$

then the expected squared error is bounded as follows:

$$\mathbb{E}\{\|\overline{\boldsymbol{w}}'_n - w^*\|^2\} \leq \left(1 - \frac{\gamma\mu\nu}{8}\right)^n\left(\mathbb{E}\{\|\overline{\boldsymbol{w}}_0 - w^*\|^2\} + \frac{(3 - \|\mathcal{D}\|)\mathbb{E}\{\|\boldsymbol{\mathcal{W}}'_0\|^2\}}{K(1 - \|\mathcal{D}\|)}\right.$$

$$\left. + \frac{2\gamma^2\mu^2\|\nabla\mathcal{J}(\overline{\boldsymbol{\mathcal{W}}}'_0)\|^2}{K(1 - \|\mathcal{D}\|)}\right) + \frac{8\gamma}{\mu\nu K}\left(\frac{4\sigma_\mu^2\|\nabla\mathcal{J}(\mathcal{W}^*)\|^2}{K} + \theta_\mu^2\sigma_s^2\right)$$

$$+ \frac{576\gamma^2\delta\|\mathcal{D}\|^2}{\nu(1 - \|\mathcal{D}\|)}\left(\frac{3\sigma_\mu^2\|\nabla\mathcal{J}(\mathcal{W}^*)\|^2}{(1 - \|\mathcal{D}\|)K} + \theta_\mu^2\sigma_s^2\right). \tag{21}$$

*Proof:* The detailed proof can be found in the Appendix.

**Remark 3** (***Impact of protection vector and random matrix-step-sizes***) *The parameter $\tau$ does not affect steady-state performance, while $\sigma_\mu^2$ does. The impact grows with data heterogeneity (caused by $\|\nabla\mathcal{J}(\mathcal{W}^*)\|^2$ in the bounds). Choosing a smaller $\gamma$ can improve network estimation accuracy.*

**Remark 4** (***Sparsely connected network***) *In sparsely connected networks ($\|\mathcal{D}\|^2 \to 1$), the terms $O(\gamma^2\theta_\mu^2\|\mathcal{D}\|^2/(1 - \|\mathcal{D}\|)^2)$ in DSG-RMS and $O(\gamma^2\sigma_\mu^2\|\mathcal{D}\|^2/(1 - \|\mathcal{D}\|)^2)$ in EDSG-RMS strongly affect steady-state performance. Since $\theta_\mu^2 > \sigma_\mu^2$, EDSG-RMS can achieve better steady-state performance than DSG-RMS.*

Define the mutual information $I(X; Y)$, which measures the amount of information learned about $X$ by observing $Y$ Cover (1999); Li et al. (2020). Then, we have privacy-preserving results for network estimation and gradients.

**Theorem 3 (Privacy of Network Estimates and Gradients)** *Consider the setting where all random step-sizes share the same variance $\sigma_\mu^2$. Then, $\lim_{\sigma_\mu^2 \to \infty} I(S_k(n); Z_k(n)) = 0$, where*

*$S_k(n) = \widehat{\nabla J}_k\big(\boldsymbol{w}_k(n-1); \boldsymbol{x}_k(n)\big), Z_k(n) = \boldsymbol{M}_k(n)\widehat{\nabla J}_k\big(\boldsymbol{w}_k(n-1); \boldsymbol{x}_k(n)\big)$, and $\boldsymbol{M}_k(n)$ denotes the random step-size applied by agent $k$ at time $n$. In addition, for an external eavesdropper, even upon observing a large number of intermediate variables $\boldsymbol{\psi}_k^c(n)$, the eavesdropper remains unable to reconstruct the network's estimate without knowledge of $\tau$.*

*Proof: By computing the mutual information between the private gradient and the observed signal under multiplicative noise, the desired result follows directly. The proof proceeds analogously to the analysis of additive noise perturbation in Li et al. (2020).*

**Remark 5** (***Trade-off between privacy and algorithm convergence speed***) *According to Theorem 3, gradient information leakage can be effectively suppressed by choosing a sufficiently large variance for the random step-sizes. However, Theorems 1 and 2 indicate that increasing the variance degrades the network's estimation accuracy. Fortunately, this degradation can be alleviated by reducing the parameter $\gamma$. Nevertheless, a smaller $\gamma$ leads to slower convergence of the algorithm, revealing an inherent trade-off between privacy preservation and algorithm convergence speed. The trade-off can be alleviated, by adopting a time-varying strategy $\gamma(k)$ that gradually decreases over iterations.*

## 4.3 Low-Complexity Choice of Matrix-Step-Size

When all elements of $\boldsymbol{M}_k(n)$ are non-zero, computing $\boldsymbol{M}_k(n)\widehat{\nabla J}_k(\boldsymbol{w}_k(n-1); \boldsymbol{x}_k(n))$ requires $O(L^2)$ operations. To reduce this cost, we propose two sparse alternatives for $\boldsymbol{M}_k(n)$:

- Upper Triangular Structure: $\boldsymbol{M}_k(n)$ is constrained to the following upper triangular form:

$$\boldsymbol{M}_k(n) = \begin{bmatrix} \boldsymbol{\mu}_{k1}(n) & \star_1 & 0 & 0 \\ 0 & \boldsymbol{\mu}_{k2}(n) & \star_2 & \vdots \\ \vdots & \vdots & \ddots & \star_{L-1} \\ 0 & 0 & 0 & \boldsymbol{\mu}_{kL}(n) \end{bmatrix}, \tag{22}$$

  where $\star_\ell$ represents non-zero elements;

- Sparse Randomized Structure: In addition to the diagonal elements, $L$ off-diagonal entries are randomly selected and assigned values drawn from zero-mean random variables.

In both cases, the complexity of computing $\boldsymbol{M}_k(n)\widehat{\nabla J}_k(\boldsymbol{w}_k(n-1); \boldsymbol{x}_k(n))$ is reduced to $O(L)$.

## 5 Experimental Verification

In all algorithms, the initial estimates are drawn uniformly from $[-1,1]$. The network includes five agents with randomly generated links satisfying Assumption 1, and combination matrices are built using the Laplacian rule. Performance is measured by the mean-square deviation (MSD) and the squared gradient norm: $\text{MSD}(n) = \frac{1}{K}\sum_{k=1}^K \|\boldsymbol{w}_k(n) - w^*\|^2$, $\overline{\|\nabla J\|^2}(n) = \frac{1}{n}\sum_{\ell=1}^n \|\nabla J(\overline{\boldsymbol{w}}_{\ell-1})\|^2$.

### 5.1 Application: Adaptive Filtering

We consider a linear adaptive filtering task where each agent observes streaming data $\boldsymbol{d}_k(n) = \boldsymbol{x}_k^\mathsf{T}(n)w_k^o + \boldsymbol{v}_k(n)$, $k = 1, 2, ..., K$, with local optimum $w_k^o$ and zero-mean noise $\boldsymbol{v}_k(n)$. For Gaussian noise $\boldsymbol{v}_k(n)$, the global optimum of the network MSE cost, $\min \frac{1}{2K}\sum_{k=1}^K \mathbb{E}\{(\boldsymbol{d}_k(n) - \boldsymbol{x}_k^\mathsf{T}(n)w)^2\}$, is $w^* = (\sum_{k=1}^K R_{x,k})^{-1}(\sum_{k=1}^K R_{x,k}w_k^o)$, where $R_{x,k} = \mathbb{E}\{\boldsymbol{x}_k(n)\boldsymbol{x}_k^\mathsf{T}(n)\}$. In the simulation, $\boldsymbol{x}_k(n) \sim \mathcal{N}(0, \sigma_{x,k}^2\boldsymbol{I}_5)$, $\boldsymbol{v}_k(n) \sim \mathcal{N}(0, \sigma_{v,k}^2)$, and $w_k^o \sim \mathcal{N}(0, \boldsymbol{I}_5)$, with $\sigma_{x,k}^2, \sigma_{v,k}^2 \sim \mathcal{U}(0,1)$. To test the tracking, $w_k^o$ changes sign during intermediate iterations. Fig. 1 shows convergence curves of DSG-RMS and EDSG-RMS under homogeneous and heterogeneous networks, using both stochastic and exact gradients. In Fig. 1(c), the notations $\boldsymbol{\psi}_{k,1}(n)$, $\boldsymbol{w}_{k,1}(n-1)$, $\boldsymbol{\psi}_{k,1}^c(n)$, and $w_1^*$ refer to the first elements of $\boldsymbol{\psi}_k(n)$, $\boldsymbol{w}_k(n-1)$, $\boldsymbol{\psi}_k^c(n)$, and $w^*$, respectively. Parameters are set to $\gamma = 0.0001$, $\mu = 1$, $\tau = 1$, and $\sigma_z^2 = 0.0001$ for DSG-RMS/EDSG-RMS. For PD-LMSRizk

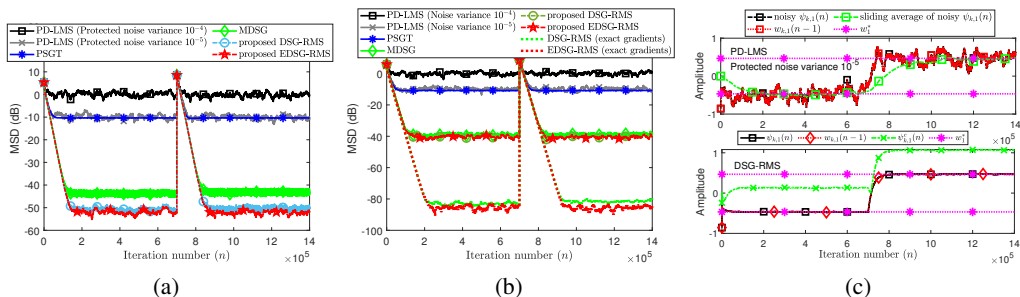

(a)  (b)  (c)

Figure 1: Convergence of DSG-RMS and EDSG-RMS compared with PD-LMS Rizk et al. (2023), PSGT Ding et al. (2021), and MDSGHan et al. (2025). (a) MSD curves (10 runs) on a homogeneous network ($w_1^o = ... = w_K^o$); (b) MSD curves (10 runs) on a heterogeneous network ($w_1^o \neq ... \neq w_K^o$); (c) Convergence curves (1 run) of $\boldsymbol{\psi}_{k,1}(n)$, $\boldsymbol{w}_{k,1}(n-1)$, and $\boldsymbol{\psi}_{k,1}^c(n)$ in Fig. 1(b) at agent $k = 1$.

et al. (2023), PSGT Ding et al. (2021), and MDSGHan et al. (2025), the protection noise variances are 0.0001, $\sqrt{0.1 \cdot 0.8^n}$, and 0.0001, respectively.[3] As a comparison, the MSD curve of PD-LMS using a protection noise variance $10^{-5}$ is included. As shown in Fig. 1, our algorithms outperform the comparison methods in terms of convergence performance, and effectively prevent external agents from inferring the network estimate through sliding averages. For PD-LMS with protection noise variance $10^{-5}$, an eavesdropper can approximate the target parameters via averaging, as seen in Fig. 1(c). When the protection noise variance is increased to $10^{-4}$, the PD-LMS fails to estimate the model parameters, under the network parameter settings. Moreover, under heterogeneous data, EDSG-RMS outperforms DSG-RMS when both use exact gradients, as illustrated in Fig. 1(b).

## 5.2 APPLICATION: DISTRIBUTED LEARNING

In the second experiment, we evaluate the proposed strategies by collaboratively training a convolutional neural network (CNN) over a random network of five agents. The MNIST dataset is evenly divided among the agents, with each missing two digit classes: agent 1 (0, 1), agent 2 (2, 3), agent 3 (4, 5), agent 4 (6, 7), and agent 5 (8, 9). The CNN consists of three convolutional layers (each with $5 \times 5$ kernels, 12 filters, and Sigmoid activations), followed by a fully connected layer that outputs 10 classes. Training uses cross-entropy loss Zhang & Sabuncu (2018), $\min -\frac{1}{K}\sum_{k=1}^{K}\frac{1}{N_k}\sum_{n=1}^{N_k}\sum_{\ell=1}^{10} y_{k\ell,n}\log(\hat{y}_{k\ell,n})$, where $N_k$ is the number of samples, and $y_{k\ell,n}, \hat{y}_{k\ell,n}$ denote the true label and predicted probability for class $\ell$ of the $n$-th sample at agent $k$. Each agent randomly selects a sample at each iteration to compute a stochastic gradient for parameter updates.Other settings include protection noise variance $\sigma_{z,k}^2 = 10^{-2}$ for DSG-RMS and EDSG-RMS, $\gamma = 0.02$, $\mu = 1$, and $\tau = 0.1$. We assume agent 1 is honest-but-curious: it follows the protocol but attempts to infer agent 2's local data using the DLG attack Zhu et al. (2019), leveraging available weight and gradient estimates. For the DSG and DSG-RMS, the estimated weights and gradient informations are $\{\boldsymbol{w}_2(n-1), (-\boldsymbol{\psi}_2(n) + \boldsymbol{w}_2(n-1))/\gamma\}$ and $\{\hat{\boldsymbol{w}}_2(n-1), (-\boldsymbol{\psi}_2^c(n-1) + \frac{\tau}{\sqrt{L}}\|\boldsymbol{w}_1(n-1)\| \cdot \mathbb{1}_L + \hat{\boldsymbol{w}}_2(n-1))/\gamma\}$, respectively, where $\hat{\boldsymbol{w}}_2(n) \approx \sum_{\ell \in \mathcal{N}_2} a_{\ell 2}\boldsymbol{\psi}_\ell^c(n) - \frac{\tau}{\sqrt{L}}\|\boldsymbol{w}_1(n-1)\| \cdot \mathbb{1}_L$. Due to the interplay between the correction, protection, and combination steps in the EDSG-RMS, agent 1 is unable to estimate $\boldsymbol{\psi}_2'(n)$ when the initial values $\boldsymbol{c}_2'(-1)$ and $\boldsymbol{w}_2'(0)$ are randomly selected. In this case, the agent lacks access to the required gradient information. To test the protective role of the random matrix, we applied $\boldsymbol{w}_2'(n-1)$ and $\boldsymbol{M}_2(n)\widehat{\nabla J}_2(\boldsymbol{w}_2'(n-1); \boldsymbol{x}_2(n))$ in DLG diagnosis under the EDSG-RMS. As shown in Fig. 2(a), the squared gradient norm and prediction accuracy of the proposed methods are comparable to DSG. Fig. 2(b) demonstrates that our methods are more effective at mitigating DLG attacks.

## 5.3 APPLICATION: TARGET LOCALIZATION

Let the unknown target location in the Cartesian plane be $w^* = [w_1^*, w_2^*]^\top$. Four anchor agents at $p_k = [x_k, y_k]^\top$, $k = 1, 2, 3, 4$, obtain noisy measurements of the distance $\boldsymbol{r}_k(n)$ and direction

---

[3]PD-LMS here is a local differential privacy version, where the noise is added to each agent.

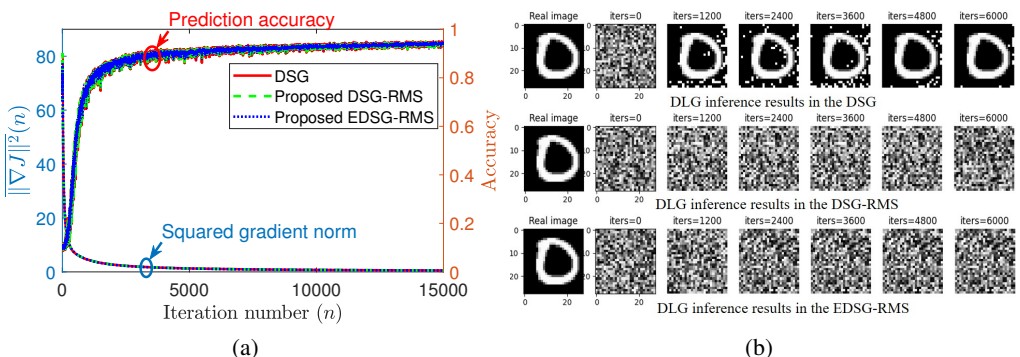

(a)  (b)

Figure 2: (a) Convergence curves (1 run) of DSG-RMS and EDSG-RMS with $\gamma = 0.02$; (b) Inference results from the DLG.

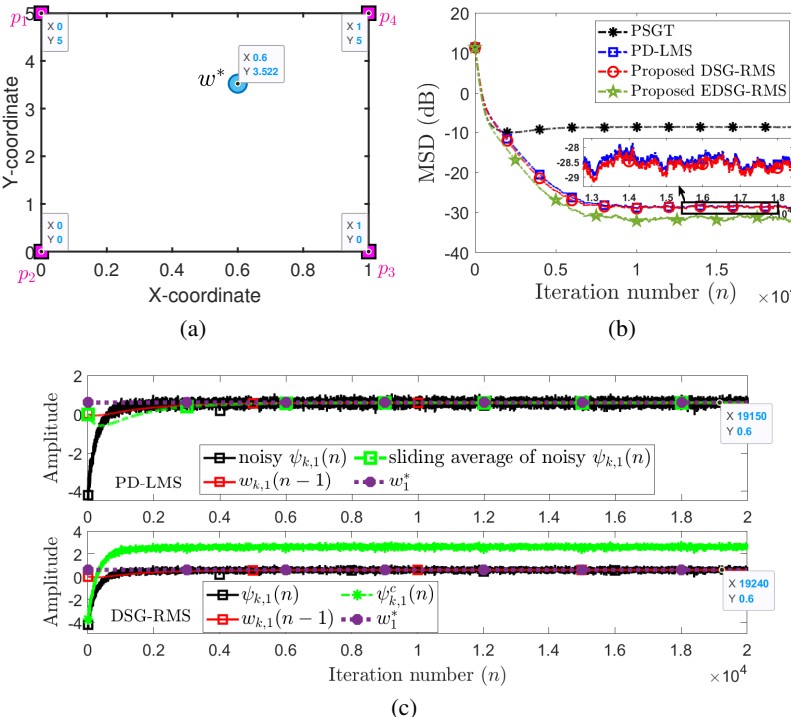

(a)  (b)

(c)

Figure 3: Convergence curves (100 runs) in target localization task. (a) localization of target and anchor agents; (b) MSD curves; (c) The changes in specific variables at agent $k = 1$.

$z_k(n)$ to the target at time $n$. The localization model in Sayed (2014) expresses the relationship as: $d_k(n) = r_k(n) + z_k^\top(n)p_k = z_k^\top(n)w^* + v_k(n), k = 1, 2, 3, 4$, where $v_k(n)$ is zero-mean Gaussian noise. To estimate $w^*$, agents share the estimated values with neighbors. Direct sharing, however, risks exposing location information. To address this, EDSG-RMS and DSG-RMS are applied. The results, shown in Figs. 3(b) and (c), were obtained with parameters $\gamma = 4$, $\mu = 1$, $\tau = 0.8$, and $\sigma_z^2 = 0.001$. As observed, the EDSG-RMS algorithm slightly outperforms the others. The DSG-RMS and PD-LMS algorithms achieve similar performance, but the DSG-RMS and EDSG-RMS provide better privacy protection. This is because the sliding average result of the transmitted values in the PD-LMS can approximate the target location information, shown in Fig. 3(c).

## 6 CONCLUSION

This paper has introduced two privacy-preserving decentralized learning algorithms, DSG-RMS and EDSG-RMS, designed to mitigate information leakage in both network-estimated values and local

gradients/data. We have also established their convergence guarantees for convex objectives, explicit accounting for the network estimation accuracy and privacy-preserving effects of non-zero protection vectors and random matrix-step-sizes. Our analysis reveals a fundamental trade-off: while increasing the variance of the step-size enhances gradient privacy, it inevitably degrades network estimation accuracy. However, this degradation can be effectively mitigated by reducing the algorithmic parameter $\gamma$. Finally, applications in distributed filtering, learning, and target localization demonstrate the effectiveness of these algorithms, highlighting their practical value in privacy-sensitive optimization.

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
