## PROOF OF THEOREM 1

### A.1 DECOUPLED/TRANSFORMED WEIGHT UPDATES OF DSG-RMS

To facilitate subsequent theoretical analysis, we presents the compact equivalent update form of the DSG-RMS algorithm.

Define the network variables:

$$\boldsymbol{\mathcal{W}}_n \triangleq \mathrm{col}\{\boldsymbol{w}_1(n), \boldsymbol{w}_2(n), ..., \boldsymbol{w}_K(n)\}, \tag{S1}$$

$$\boldsymbol{\mathcal{W}}'_n \triangleq \mathrm{col}\{\boldsymbol{w}'_1(n), \boldsymbol{w}'_2(n), ..., \boldsymbol{w}'_K(n)\}, \tag{S2}$$

as well as network gradient noise $\boldsymbol{\mathcal{S}}_n$, $\boldsymbol{\mathcal{S}}'_n$, network random matrix $\boldsymbol{\mathcal{M}}_n$, and vectors $\boldsymbol{\mathcal{C}}_{n-1}$ and $\boldsymbol{\mathcal{C}}'_{n-1}$:

$$\boldsymbol{\mathcal{S}}_n = \mathrm{col}\{\boldsymbol{s}_{1,n}(\boldsymbol{w}_1(n-1)), ..., \boldsymbol{s}_{K,n}(\boldsymbol{w}_K(n-1))\}, \tag{S3}$$

$$\boldsymbol{\mathcal{S}}'_n = \mathrm{col}\{\boldsymbol{s}_{1,n}(\boldsymbol{w}'_1(n-1)), ..., \boldsymbol{s}_{K,n}(\boldsymbol{w}'_K(n-1))\}, \tag{S4}$$

$$\boldsymbol{\mathcal{M}}_n = \mathrm{blkdiag}\{\boldsymbol{M}_1(n), \boldsymbol{M}_2(n), ..., \boldsymbol{M}_K(n)\}, \tag{S5}$$

$$\boldsymbol{\mathcal{C}}_{n-1} = \mathrm{col}\{\|\boldsymbol{w}_1(n-1)\| \cdot \mathbb{1}_L, ..., \|\boldsymbol{w}_K(n-1)\| \cdot \mathbb{1}_L\}, \tag{S6}$$

$$\boldsymbol{\mathcal{C}}'_{n-1} = \mathrm{col}\{\|\boldsymbol{w}'_1(n-1)\| \cdot \mathbb{1}_L, ..., \|\boldsymbol{w}'_K(n-1)\| \cdot \mathbb{1}_L\}, \tag{S7}$$

where gradient noise $\boldsymbol{s}_{k,n}(\boldsymbol{w}_k(n-1)) = \widehat{\nabla J}_k(\boldsymbol{w}_k(n-1); \boldsymbol{x}_k(n)) - \nabla J_k(\boldsymbol{w}_k(n-1))$.

The update rule given by (5) can then be reformulated as:

$$\boldsymbol{\mathcal{W}}_n = \mathcal{A}\boldsymbol{\mathcal{W}}_{n-1} - \mathcal{A}\gamma\mu\nabla\mathcal{J}(\boldsymbol{\mathcal{W}}_{n-1}) + \mathcal{A}\gamma(\mu\boldsymbol{I}_{KL} - \boldsymbol{\mathcal{M}}_n)$$
$$\times \nabla\mathcal{J}(\boldsymbol{\mathcal{W}}_{n-1}) - \mathcal{A}\gamma\boldsymbol{\mathcal{M}}_n\boldsymbol{\mathcal{S}}_n + \frac{\tau}{\sqrt{L}}(\mathcal{A} - \boldsymbol{I}_{KL})(\boldsymbol{\mathcal{C}}_{n-1} - \overline{\boldsymbol{\mathcal{C}}}_{n-1}), \tag{S8}$$

where $\mathcal{A} = A \otimes \boldsymbol{I}_L$, $\nabla\mathcal{J}(\boldsymbol{\mathcal{W}}_{n-1}) = \mathrm{col}\{\nabla J_k(\boldsymbol{w}_k(n-1))\}_{k=1}^K$ and $\overline{\boldsymbol{\mathcal{C}}}_n = \|\frac{1}{K}\sum_{k=1}^K \boldsymbol{w}_k(n)\| \cdot \mathbb{1}_{KL}$.

Following the analytical approach in Zhao & Sayed (2014), we proceed to derive the transformed weight updates of (5). Using the structure of $\mathcal{X}^{-1}$ in (7), we define $\overline{\boldsymbol{w}}_n$, and $\breve{\boldsymbol{w}}_n$ as

$$\begin{bmatrix} \overline{\boldsymbol{w}}_n \\ \breve{\boldsymbol{w}}_n \end{bmatrix} = \begin{bmatrix} \Gamma^{\mathsf{T}} \\ \frac{1}{c}\mathcal{X}_L \end{bmatrix} \boldsymbol{\mathcal{W}}_n, \tag{S9}$$

where $\overline{\boldsymbol{w}}_n = \frac{1}{K}\sum_{k=1}^K \boldsymbol{w}_k(n)$. From (S9), we have $\boldsymbol{\mathcal{W}}_n - \overline{\boldsymbol{\mathcal{W}}}_n = c\mathcal{X}_R\breve{\boldsymbol{w}}_n$ with $\overline{\boldsymbol{\mathcal{W}}}_n = (\frac{1}{K}\mathbb{1}_K\mathbb{1}_K^{\mathsf{T}} \otimes \boldsymbol{I}_L)\boldsymbol{\mathcal{W}}_n$.

By left-multiplying both sides of (S8) with $\mathcal{X}^{-1}$, we obtain the following transformed update formulas of the DSG-RMS:

$$\overline{\boldsymbol{w}}_n = \overline{\boldsymbol{w}}_{n-1} - \gamma\mu\Gamma^{\mathsf{T}}\nabla\mathcal{J}(\boldsymbol{\mathcal{W}}_{n-1}) + \gamma\Gamma^{\mathsf{T}}(\mu\boldsymbol{I}_{KL} - \boldsymbol{\mathcal{M}}_n)$$
$$\times \nabla\mathcal{J}(\boldsymbol{\mathcal{W}}_{n-1}) - \gamma\Gamma^{\mathsf{T}}\boldsymbol{\mathcal{M}}_n\boldsymbol{\mathcal{S}}_n, \tag{S10}$$

and

$$\breve{\boldsymbol{w}}_n = \mathcal{D}\breve{\boldsymbol{w}}_{n-1} - \gamma\mu\mathcal{D}\frac{1}{c}\mathcal{X}_L\nabla\mathcal{J}(\boldsymbol{\mathcal{W}}_{n-1}) + \mathcal{D}\frac{1}{c}\mathcal{X}_L\gamma$$
$$\times (\mu\boldsymbol{I}_{KL} - \boldsymbol{\mathcal{M}}_n)\nabla\mathcal{J}(\boldsymbol{\mathcal{W}}_{n-1}) - \mathcal{D}\frac{1}{c}\mathcal{X}_L\gamma\boldsymbol{\mathcal{M}}_n\boldsymbol{\mathcal{S}}_n$$
$$+ \frac{\tau}{\sqrt{L}}(\mathcal{D} - \boldsymbol{I}_{(K-1)L})\frac{1}{c}\mathcal{X}_L(\boldsymbol{\mathcal{C}}_{n-1} - \overline{\boldsymbol{\mathcal{C}}}_{n-1}). \tag{S11}$$

It can be observed that in (S10) and (S11), the updates of $\overline{\boldsymbol{w}}_n$ and $\breve{\boldsymbol{w}}_n$ are decoupled through the use of the matrix $\mathcal{X}^{-1}$.

### A.2 PROOF OF THEOREM 1

Define the error vector $\overline{\boldsymbol{w}}_n^e = \overline{\boldsymbol{w}}_n - w^*$. Using Assumptions 4 and 5, we have

$$\|\nabla\mathcal{J}(\overline{\boldsymbol{\mathcal{W}}}_{n-1})\|^2 \leq 2\|\nabla\mathcal{J}(\overline{\boldsymbol{\mathcal{W}}}_{n-1}) - \nabla\mathcal{J}(\mathcal{W}^*)\|^2 + 2\|\nabla\mathcal{J}(\mathcal{W}^*)\|^2$$

$$\leq 4K\delta(J(\overline{\boldsymbol{w}}_{n-1}) - J(w^*)) + 2\|\nabla\mathcal{J}(\mathcal{W}^*)\|^2, \tag{S12}$$

and

$$\mathbb{E}\{\|\Gamma^{\mathsf{T}}(\mu\boldsymbol{I}_{KL} - \boldsymbol{\mathcal{M}}_n)\nabla\mathcal{J}(\mathcal{W}_{n-1})\|^2\}$$
$$\leq \frac{2\sigma_\mu^2\delta^2}{K^2}c^2\|\mathcal{X}_R\|^2\|\breve{\boldsymbol{w}}_{n-1}\|^2 + \frac{4\sigma_\mu^2}{K}\left(\delta^2\|\overline{\boldsymbol{w}}_{n-1}^e\|^2 + \frac{\|\nabla\mathcal{J}(\mathcal{W}^*)\|^2}{K}\right), \tag{S13}$$

where $\|\nabla J_k(\overline{\boldsymbol{w}}_{n-1}) - \nabla J_k(w^*)\|^2 \leq 2\delta(J_k(\overline{\boldsymbol{w}}_{n-1}) - J_k(w^*))$ and $\|\nabla J_k(\overline{\boldsymbol{w}}_{n-1}) - \nabla J_k(w^*)\|^2 \leq \delta^2\|\overline{\boldsymbol{w}}_{n-1} - w^*\|^2$ are used in (S12) and (S13), respectively.

From (S10), it holds that

$$\mathbb{E}\{\|\overline{\boldsymbol{w}}_n^e\|^2|\boldsymbol{\mathcal{F}}_{n-1}'\} = \|\overline{\boldsymbol{w}}_{n-1}^e - \gamma\mu\Gamma^{\mathsf{T}}\nabla\mathcal{J}(\mathcal{W}_{n-1})\|^2 + \gamma^2\mathbb{E}\{\|\Gamma^{\mathsf{T}}(\mu\boldsymbol{I}_{KL} - \boldsymbol{\mathcal{M}}_n)\nabla\mathcal{J}(\mathcal{W}_{n-1})\|^2\}$$
$$+\gamma^2\mathbb{E}\{\|\Gamma^{\mathsf{T}}\boldsymbol{\mathcal{M}}_n\boldsymbol{\mathcal{S}}_n\|^2|\boldsymbol{\mathcal{F}}_{n-1}'\}$$
$$\leq \|\overline{\boldsymbol{w}}_{n-1}^e\|^2 + \gamma^2\mu^2\Big\|\frac{1}{K}\sum_{k=1}^K\nabla J_k(\boldsymbol{w}_k(n-1))\Big\|^2 - 2\gamma\mu\Big\langle\overline{\boldsymbol{w}}_{n-1}^e, \frac{1}{K}\sum_{k=1}^K\nabla J_k(\boldsymbol{w}_k(n-1))\Big\rangle$$
$$+\gamma^2\frac{2\sigma_\mu^2\delta^2}{K^2}c^2\|\mathcal{X}_R\|^2\|\breve{\boldsymbol{w}}_{n-1}\|^2 + \gamma^2\frac{4\sigma_\mu^2}{K}\left(\delta^2\|\overline{\boldsymbol{w}}_{n-1}^e\|^2 + \frac{\|\nabla\mathcal{J}(\mathcal{W}^*)\|^2}{K}\right) + \gamma^2\frac{\theta_\mu^2\sigma_s^2}{K}, \tag{S14}$$

where (S12) and (S13) are used. Then, we apply the inequality:

$$\langle z-y, \nabla J(x)\rangle \geq J(z) - J(y) + \frac{\nu}{4}\|y-z\|^2 - \delta\|z-x\|^2, \tag{S15}$$

where $x, y, z \in \mathbb{R}^L$ and $J(z)$ is $\delta$-smooth and $\nu$-strongly convex. Setting $z = \overline{\boldsymbol{w}}_{n-1}$, $y = w^*$, and $x = \boldsymbol{w}_k(n-1)$, we get:

$$\Big\langle\overline{\boldsymbol{w}}_{n-1}-w^*, \frac{1}{K}\sum_{k=1}^K\nabla J_k(\boldsymbol{w}_k(n-1))\Big\rangle \geq J(\overline{\boldsymbol{w}}_{n-1}) - J(w^*)$$
$$+ \frac{\nu}{4}\|\overline{\boldsymbol{w}}_{n-1}^e\|^2 - \frac{\delta}{K}\|\mathcal{W}_{n-1} - \overline{\mathcal{W}}_{n-1}\|^2. \tag{S16}$$

Applying the smoothness of $J_k(\cdot)$, we obtain:

$$\Big\|\frac{1}{K}\sum_{k=1}^K\nabla J_k(\boldsymbol{w}_k(n-1))\Big\|^2$$
$$\leq \frac{2\delta^2}{K}\|\mathcal{W}_{n-1} - \overline{\mathcal{W}}_{n-1}\|^2 + 2\|\nabla J(\overline{\boldsymbol{w}}_{n-1})\|^2$$
$$\leq \frac{2\delta^2}{K}c^2\|\mathcal{X}_R\|^2\|\breve{\boldsymbol{w}}_{n-1}\|^2 + 2\|\nabla J(\overline{\boldsymbol{w}}_{n-1})\|^2. \tag{S17}$$

Substituting these results back into the conditional expectation formula (S14) and provides

$$\mathbb{E}\{\|\overline{\boldsymbol{w}}_n^e|\boldsymbol{\mathcal{F}}_{n-1}\|^2\} \leq (1 - \gamma\mu\frac{\nu}{2})\|\overline{\boldsymbol{w}}_{n-1}^e\|^2 - 2\gamma\mu(J(\overline{\boldsymbol{w}}_{n-1}) - J(w^*))$$
$$+ \gamma\Big(2\mu\delta + 2\gamma\delta^2\mu^2 + \gamma\frac{2\sigma_\mu^2\delta^2}{K}\Big)\|\mathcal{X}_L\|^2\|\mathcal{X}_R\|^2\|\breve{\boldsymbol{w}}_{n-1}\|^2 + \gamma^2\Big(2\mu^2\|\nabla J(\overline{\boldsymbol{w}}_{n-1})\|^2$$
$$+ \frac{4\sigma_\mu^2\|\nabla\mathcal{J}(\mathcal{W}^*)\|^2}{K^2}\Big) + \gamma^2\frac{4\sigma_\mu^2\delta^2}{K}\|\overline{\boldsymbol{w}}_{n-1}^e\|^2 + \gamma^2\frac{\theta_\mu^2\sigma_s^2}{K}, \tag{S18}$$

where $c^2 = K\|\mathcal{X}_L\|^2$ is used again. When $\gamma$ satisfies the following conditions:

$$2\mu\delta + 2\gamma\delta^2\mu^2 + \gamma\frac{2\sigma_\mu^2\delta^2}{K} \leq 3\mu\delta, \tag{S19}$$

$$1 - \gamma\mu\frac{\nu}{2} + \gamma^2\frac{4\sigma_\mu^2\delta^2}{K} \leq 1 - \gamma\mu\frac{\nu}{4}, \ (\nu > 0) \tag{S20}$$

and

$$2\gamma\mu - 4\gamma^2\delta\mu^2 \geq \gamma\mu, \tag{S21}$$

along with using the inequality under Assumption 4:

$$\|\nabla J(\overline{\boldsymbol{w}}_{n-1})\|^2 \leq 2\delta(J(\overline{\boldsymbol{w}}_{n-1}) - J(w^*)), \tag{S22}$$

from (S18), we obtain the following results:

- (Case I: $\nu = 0$)

$$\mathbb{E}\{\|\overline{\boldsymbol{w}}_n^e\|^2\} \leq \mathbb{E}\{\|\overline{\boldsymbol{w}}_{n-1}^e\|^2\} - \gamma\mu\big(\mathbb{E}\{J(\overline{\boldsymbol{w}}_{n-1})\} - J(w^*)\big)$$
$$+3\gamma\mu\delta\|\mathcal{X}_L\|^2\|\mathcal{X}_R\|^2\mathbb{E}\{\|\breve{\boldsymbol{w}}_{n-1}\|^2\}$$
$$+\gamma^2\frac{4\sigma_\mu^2\|\nabla\mathcal{J}(\mathcal{W}^*)\|^2}{K^2} + \gamma^2\frac{\theta_\mu^2\sigma_s^2}{K}. \tag{S23}$$

- (Case II: $\nu \neq 0$)

$$\mathbb{E}\{\|\overline{\boldsymbol{w}}_n^e\|^2\} \leq (1 - \frac{\gamma\mu\nu}{4})\mathbb{E}\{\|\overline{\boldsymbol{w}}_{n-1}^e\|^2\} - \gamma\mu\big(\mathbb{E}\{J(\overline{\boldsymbol{w}}_{n-1})\} - J(w^*)\big)$$
$$+3\gamma\mu\delta\|\mathcal{X}_L\|^2\|\mathcal{X}_R\|^2\mathbb{E}\{\|\breve{\boldsymbol{w}}_{n-1}\|^2\} + \gamma^2\frac{4\sigma_\mu^2\|\nabla\mathcal{J}(\mathcal{W}^*)\|^2}{K^2}$$
$$+\gamma^2\frac{\theta_\mu^2\sigma_s^2}{K}. \tag{S24}$$

For the case I, from (S23), we have

$$\frac{\gamma\mu}{N}\sum_{n=1}^{N}\big(\mathbb{E}\{J(\overline{\boldsymbol{w}}_{n-1})\} - J(w^*)\big) \leq \frac{\mathbb{E}\{\|\overline{\boldsymbol{w}}_0^e\|^2\}}{N} + 3\gamma\mu\delta\|\mathcal{X}_L\|^2\|\mathcal{X}_R\|^2\frac{1}{N}\sum_{n=1}^{N}\mathbb{E}\{\|\breve{\boldsymbol{w}}_{n-1}\|^2\}$$
$$+\gamma^2\frac{4\sigma_\mu^2\|\nabla\mathcal{J}(\mathcal{W}^*)\|^2}{K^2} + \gamma^2\frac{\theta_\mu^2\sigma_s^2}{K}. \tag{S25}$$

Using Jensen's inequality and (S12), from (S11), we know that

$$\mathbb{E}\{\|\breve{\boldsymbol{w}}_n\|^2\} \leq \frac{1+\|\mathcal{D}\|}{2}\mathbb{E}\{\|\breve{\boldsymbol{w}}_{n-1}\|^2\} + \left(\frac{8\|\nabla\mathcal{J}(\mathcal{W}^*)\|^2}{(1-\|\mathcal{D}\|)K} + \sigma_s^2\right)$$
$$\times\gamma^2\theta_\mu^2\|\mathcal{D}\|^2 + \frac{16\gamma^2\delta\theta_\mu^2\|\mathcal{D}\|^2}{1-\|\mathcal{D}\|}(\mathbb{E}\{J(\overline{\boldsymbol{w}}_{n-1})\} - J(w^*)), \tag{S26}$$

and

$$\frac{1}{N}\sum_{n=1}^{N}\mathbb{E}\{\|\breve{\boldsymbol{w}}_n\|^2\} \leq \frac{1}{N}\frac{1+\|\mathcal{D}\|}{1-\|\mathcal{D}\|}\mathbb{E}\{\|\breve{\boldsymbol{w}}_0\|^2\} + \frac{2}{1-\|\mathcal{D}\|}\left(\frac{8\|\nabla\mathcal{J}(\mathcal{W}^*)\|^2}{(1-\|\mathcal{D}\|)K} + \sigma_s^2\right)\gamma^2\theta_\mu^2\|\mathcal{D}\|^2$$
$$+\frac{32\delta\gamma^2\theta_\mu^2\|\mathcal{D}\|^2}{(1-\|\mathcal{D}\|)^2}\frac{1}{N}\sum_{n=0}^{N-1}(\mathbb{E}\{J(\overline{\boldsymbol{w}}_n)\} - J(w^*)), \tag{S27}$$

where the following conditions (S28) and (S29) are required,

$$|\tau| \leq \frac{1-\|\mathcal{D}\|}{\sqrt{8}\|\mathcal{D} - \boldsymbol{I}_{(K-1)L}\|\|\mathcal{X}_L\|\|\mathcal{X}_R\|}, \tag{S28}$$

$$\gamma \leq \frac{1-\|\mathcal{D}\|}{4\delta\theta_\mu\|\mathcal{D}\|\|\mathcal{X}_L\|\|\mathcal{X}_R\|}. \tag{S29}$$

By applying the relation (S27), the inequality (S25) becomes:

$$\gamma\mu\left(1 - \frac{96\gamma^2\delta^2\theta_\mu^2\|\mathcal{X}_L\|^2\|\mathcal{X}_R\|^2\|\mathcal{D}\|^2}{(1-\|\mathcal{D}\|)^2}\right)\mathscr{E}_N \leq \frac{\mathbb{E}\{\|\overline{\boldsymbol{w}}_0^e\|^2\}}{N} + \frac{6\gamma\mu\delta\|\mathcal{X}_L\|^2\|\mathcal{X}_R\|^2}{N(1-\|\mathcal{D}\|)}\mathbb{E}\{\|\breve{\boldsymbol{w}}_0\|^2\}$$

$$+ \frac{6\gamma^3\theta_\mu^2\mu\delta\|\mathcal{X}_L\|^2\|\mathcal{X}_R\|^2}{1-\|\mathcal{D}\|}\left(\frac{8\|\nabla\mathcal{J}(\mathcal{W}^*)\|^2}{(1-\|\mathcal{D}\|)K}+\sigma_s^2\right)\|\mathcal{D}\|^2$$

$$+ \gamma^2\frac{4\sigma_\mu^2\|\nabla\mathcal{J}(\mathcal{W}^*)\|^2}{K^2}+\gamma^2\frac{\theta_\mu^2\sigma_s^2}{K}, \tag{S30}$$

where $\mathscr{E}_N = \frac{1}{N}\sum_{n=1}^{N}\left(\mathbb{E}\{J(\overline{\boldsymbol{w}}_{n-1})\}-J(w^*)\right)$. Under the condition

$$\frac{96\gamma^2\delta^2\theta_\mu^2\|\mathcal{X}_L\|^2\|\mathcal{X}_R\|^2\|\mathcal{D}\|^2}{(1-\|\mathcal{D}\|)^2}\leq\frac{1}{2}, \tag{S31}$$

and using the fact that $\|\breve{\boldsymbol{w}}_0\|^2 \leq \frac{1}{K}\|\boldsymbol{\mathcal{W}}_0\|^2$, the desired result (14) is achieved. By integrating conditions (S19), (S21), and (S31), we obtain (12).

For the case II (strongly-convex), we have

$$\mathbb{E}\{\|\breve{\boldsymbol{w}}_n\|^2\}\leq\frac{1+\|\mathcal{D}\|}{2}\|\breve{\boldsymbol{w}}_{n-1}\|^2+\left(\frac{8\|\nabla\mathcal{J}(\mathcal{W}^*)\|^2}{(1-\|\mathcal{D}\|)K}+\sigma_s^2\right)\gamma^2\theta_\mu^2\|\mathcal{D}\|^2$$

$$+ \frac{8\gamma^2\delta^2\theta_\mu^2\|\mathcal{D}\|^2}{1-\|\mathcal{D}\|}\mathbb{E}\{\|\overline{\boldsymbol{w}}_{n-1}^e\|^2\}, \tag{S32}$$

where the condition $\|\nabla J(\overline{\boldsymbol{w}}_{n-1})\|^2 \leq \delta^2\|\overline{\boldsymbol{w}}_{n-1}-w^*\|^2$ is used. Based on this, and combining with (S24), we derive the following relationship:

$$\underbrace{\begin{bmatrix}\mathbb{E}\{\|\overline{\boldsymbol{w}}_n^e\|^2\}\\\mathbb{E}\{\|\breve{\boldsymbol{w}}_n\|^2\}\end{bmatrix}}_{t_n}\preceq\underbrace{\begin{bmatrix}1-\frac{1}{4}\gamma\mu\nu, & 3\gamma\mu\delta\|\mathcal{X}_L\|^2\|\mathcal{X}_R\|^2\\\frac{8\gamma^2\theta_\mu^2\delta^2\|\mathcal{D}\|^2}{1-\|\mathcal{D}\|}, & \frac{1+\|\mathcal{D}\|}{2}\end{bmatrix}}_{\mathcal{Q}}$$

$$\times\begin{bmatrix}\mathbb{E}\{\|\overline{\boldsymbol{w}}_{n-1}^e\|^2\}\\\mathbb{E}\{\|\breve{\boldsymbol{w}}_{n-1}\|^2\}\end{bmatrix}+\begin{bmatrix}\gamma^2\frac{4\sigma_\mu^2\|\nabla\mathcal{J}(\mathcal{W}^*)\|^2}{K^2}+\gamma^2\frac{\theta_\mu^2\sigma_s^2}{K}\\\left(\frac{8\|\nabla\mathcal{J}(\mathcal{W}^*)\|^2}{(1-\|\mathcal{D}\|)K}+\sigma_s^2\right)\gamma^2\theta_\mu^2\|\mathcal{D}\|^2\end{bmatrix}. \tag{S33}$$

Iterating (S33), we obtain:

$$t_n\preceq\mathcal{Q}^n t_0+(\boldsymbol{I}_2-\mathcal{Q})^{-1}\begin{bmatrix}\gamma^2\frac{4\sigma_\mu^2\|\nabla\mathcal{J}(\mathcal{W}^*)\|^2}{K^2}+\gamma^2\frac{\theta_\mu^2\sigma_s^2}{K}\\\left(\frac{8\|\nabla\mathcal{J}(\mathcal{W}^*)\|^2}{(1-\|\mathcal{D}\|)K}+\sigma_s^2\right)\gamma^2\theta_\mu^2\|\mathcal{D}\|^2\end{bmatrix}$$

$$\preceq\left(1-\frac{\gamma\mu\nu}{4}\right)^n\|t_0\|_1\cdot\mathbb{1}_2+(\boldsymbol{I}_2-\mathcal{Q})^{-1}\begin{bmatrix}\gamma^2\frac{4\sigma_\mu^2\|\nabla\mathcal{J}(\mathcal{W}^*)\|^2}{K^2}+\gamma^2\frac{\theta_\mu^2\sigma_s^2}{K}\\\left(\frac{8\|\nabla\mathcal{J}(\mathcal{W}^*)\|^2}{(1-\|\mathcal{D}\|)K}+\sigma_s^2\right)\gamma^2\theta_\mu^2\|\mathcal{D}\|^2\end{bmatrix}, \tag{S34}$$

where the second inequality follows from the condition $\|\mathcal{Q}\|_1 = \max\left\{1-\frac{1}{4}\gamma\mu\nu+\frac{8\gamma^2\theta_\mu^2\delta^2\|\mathcal{D}\|^2}{1-\|\mathcal{D}\|}, 3\gamma\mu\delta\|\mathcal{X}_L\|^2\|\mathcal{X}_R\|^2+\frac{1+\|\mathcal{D}\|}{2}\right\}\leq 1-\frac{1}{8}\gamma\mu\nu$, which holds under the condition:

$$\gamma\leq\min\left\{\frac{\mu\nu(1-\|\mathcal{D}\|)}{64\theta_\mu^2\delta^2\|\mathcal{D}\|^2}, \frac{4(1-\|\mathcal{D}\|)}{(24\delta\|\mathcal{X}_L\|^2\|\mathcal{X}_R\|^2+\nu)\mu}\right\}. \tag{S35}$$

Next, we observe that:

$$(\boldsymbol{I}_2-\mathcal{Q})^{-1}\preceq\frac{16}{\gamma\mu\nu(1-\|\mathcal{D}\|)}\begin{bmatrix}\frac{1-\|\mathcal{D}\|}{2}, & 3\gamma\mu\delta\|\mathcal{X}_L\|^2\|\mathcal{X}_R\|^2\\\frac{8\gamma^2\theta_\mu^2\delta^2\|\mathcal{D}\|^2}{1-\|\mathcal{D}\|}, & \frac{1}{4}\gamma\mu\nu\end{bmatrix}, \tag{S36}$$

which requires

$$\gamma\leq\frac{(1-\|\mathcal{D}\|)}{8\sqrt{6}\delta\theta_\mu\|\mathcal{X}_L\|\|\mathcal{X}_R\|\|\mathcal{D}\|}\sqrt{\frac{\nu}{\delta}}. \tag{S37}$$

Finally, substituting (S36) into (S34) yields the desired result.

## PROOF OF THEOREM 2

### B.1 DECOUPLED/TRANSFORMED WEIGHT UPDATES OF EDSG-RMS

Building on analytical frameworks in Alghunaim & Yuan (2022); Zhao & Sayed (2014), we introduce an auxiliary (dual) network variable $\mathcal{R}'_n = \mathrm{col}\{r'_k(n)\}_{k=1}^K$, where each $r'_k(n)$ is an $L \times 1$ vector. The update rule (6) then admits the equivalent primal-dual representation:

$$\begin{cases} \mathcal{W}'_n = (2\mathcal{A} - \boldsymbol{I}_{KL})\mathcal{W}'_{n-1} - \mathcal{A}\gamma\mu\nabla\mathcal{J}(\mathcal{W}'_{n-1}) - \mathcal{R}'_{n-1} \\ \qquad + \mathcal{A}\gamma\mu\nabla\mathcal{J}(\overline{\mathcal{W}}'_{n-1}) - \mathcal{A}\gamma\boldsymbol{\mathcal{M}}_n\boldsymbol{\mathcal{S}}'_n \\ \qquad + \mathcal{A}\gamma(\mu\boldsymbol{I} - \boldsymbol{\mathcal{M}}_n)\nabla\mathcal{J}(\mathcal{W}'_{n-1}), \qquad\qquad\qquad\text{(S38a)} \\ \mathcal{R}'_n = \mathcal{R}'_{n-1} - \mathcal{A}\gamma\mu\nabla\mathcal{J}(\overline{\mathcal{W}}'_{n-1}) + \mathcal{A}\gamma\mu\nabla\mathcal{J}(\overline{\mathcal{W}}'_n) \\ \qquad + (\boldsymbol{I} - \mathcal{A})\mathcal{W}'_{n-1} + \dfrac{\tau}{\sqrt{L}}(\boldsymbol{I}_{KL} - \mathcal{A}) \\ \qquad \times (\mathcal{C}'_{n-1} - \overline{\mathcal{C}}'_{n-1}), \qquad\qquad\qquad\qquad\qquad\text{(S38b)} \end{cases}$$

where $\overline{\mathcal{C}}'_n = \|\frac{1}{K}\sum_{k=1}^K w'_k(n)\| \cdot \mathbb{1}_{KL}$, $\overline{\mathcal{W}}'_n = (\frac{1}{K}\mathbb{1}_K\mathbb{1}_K^{\mathsf{T}} \otimes \boldsymbol{I}_L)\mathcal{W}'_n$, and initial value $\mathcal{R}'_0 = \mathcal{A}\gamma\mu\nabla\mathcal{J}(\overline{\mathcal{W}}'_0) - (\boldsymbol{I} - \mathcal{A})\mathcal{W}'_0$.

Define $\overline{w}'_n, \breve{w}'_n, \overline{r}'_n$, and $\breve{r}'_n$ as

$$\begin{bmatrix} \overline{w}'_n \\ \breve{w}'_n \end{bmatrix} = \begin{bmatrix} \Gamma^{\mathsf{T}} \\ \frac{1}{c}\mathcal{X}_L \end{bmatrix}\mathcal{W}'_n, \quad \begin{bmatrix} \overline{r}'_n \\ \breve{r}'_n \end{bmatrix} = \begin{bmatrix} \Gamma^{\mathsf{T}} \\ \frac{1}{c}\mathcal{B}^{-1}\mathcal{X}_L \end{bmatrix}\mathcal{R}'_n, \qquad\text{(S39)}$$

where $\mathcal{B}^2 = \boldsymbol{I}_{(K-1)L} - \mathcal{D}$. From (S39), we have $\mathcal{W}'_n - \overline{\mathcal{W}}'_n = c\mathcal{X}_R\breve{w}'_n$ with $\overline{\mathcal{W}}'_n = (\frac{1}{K}\mathbb{1}_K\mathbb{1}_K^{\mathsf{T}} \otimes \boldsymbol{I}_L)\mathcal{W}'_n$. By left-multiplying both sides of (S38a), and (S38b) with $[\Gamma^{\mathsf{T}}; \frac{1}{c}\mathcal{X}_L]$ and $[\Gamma^{\mathsf{T}}; \frac{1}{c}\mathcal{B}^{-1}\mathcal{X}_L]$, respectively, we obtain the following transformed update formulas of the EDSG-RMS:

$$\overline{w}'_n = \overline{w}'_{n-1} - \gamma\mu\Gamma^{\mathsf{T}}\nabla\mathcal{J}(\mathcal{W}'_{n-1}) - \gamma\Gamma^{\mathsf{T}}\boldsymbol{\mathcal{M}}_n\boldsymbol{\mathcal{S}}'_n \\ \qquad + \gamma\Gamma^{\mathsf{T}}(\mu\boldsymbol{I}_{KL} - \boldsymbol{\mathcal{M}}_n)\nabla\mathcal{J}(\mathcal{W}'_{n-1}), \qquad\text{(S40)}$$

and

$$\begin{bmatrix} \breve{w}'_n \\ \breve{r}'_n \end{bmatrix} = \mathcal{G}\begin{bmatrix} \breve{w}'_{n-1} \\ \breve{r}'_{n-1} \end{bmatrix} - \begin{bmatrix} \frac{\gamma}{c}\mathcal{D}\mathcal{X}_L\boldsymbol{\mathcal{M}}_n\boldsymbol{\mathcal{S}}'_n \\ \boldsymbol{0} \end{bmatrix} + \begin{bmatrix} \frac{\gamma}{c}\mathcal{D}\mathcal{X}_L(\mu\boldsymbol{I}_{KL} - \boldsymbol{\mathcal{M}}_n)\nabla\mathcal{J}(\mathcal{W}'_{n-1}) \\ \frac{\tau}{\sqrt{L}}\mathcal{B}\frac{1}{c}\mathcal{X}_L(\mathcal{C}'_{n-1} - \overline{\mathcal{C}}'_{n-1}) \end{bmatrix}$$
$$- \frac{\gamma\mu}{c}\begin{bmatrix} \mathcal{D}\mathcal{X}_L(\nabla\mathcal{J}(\mathcal{W}'_{n-1}) - \nabla\mathcal{J}(\overline{\mathcal{W}}'_{n-1})) \\ \mathcal{B}^{-1}\mathcal{D}\mathcal{X}_L(\nabla\mathcal{J}(\overline{\mathcal{W}}'_{n-1}) - \nabla\mathcal{J}(\overline{\mathcal{W}}'_n)) \end{bmatrix}, \qquad\text{(S41)}$$

where

$$\mathcal{G} = \begin{bmatrix} 2\mathcal{D} - \boldsymbol{I}_{(K-1)L}, & -\mathcal{B} \\ \mathcal{B}, & \boldsymbol{I}_{(K-1)L} \end{bmatrix}. \qquad\text{(S42)}$$

For the matrix $\mathcal{G}$, there exists a fundamental factorization $\mathcal{G} = VPV^{-1}$, where $P$ is a diagonal matrix with entries given by $\{\lambda_i \pm j\sqrt{\lambda_i - \lambda_i^2}, i = 2, 3, ..., K\}$.[4] We partition the matrices $V^{-1}$ and $V$ as $V^{-1} = [V_L, V_R]$ and $V = [V_U; V_D]$, where $\|V\|^2 \leq 4$ and $\|V^{-1}\|^2 \leq \sigma_b^{-1}$, with $\sigma_b = \min\{\lambda_i, i = 2, 3, ..., K\}$Alghunaim & Yuan (2022).

Then, by left-multiplying both sides of (S41) with $V^{-1}$, we have

$$\underbrace{V^{-1}\begin{bmatrix} \breve{w}'_n \\ \breve{r}'_n \end{bmatrix}}_{h_n} = Ph_{n-1} - \frac{\gamma}{c}V_L\mathcal{D}\mathcal{X}_L\boldsymbol{\mathcal{M}}_n\boldsymbol{\mathcal{S}}'_n + V^{-1}\begin{bmatrix} \frac{\gamma}{c}\mathcal{D}\mathcal{X}_L(\mu\boldsymbol{I}_{KL} - \boldsymbol{\mathcal{M}}_n)\nabla\mathcal{J}(\mathcal{W}'_{n-1}) \\ \frac{\tau}{\sqrt{L}}\mathcal{B}\mathcal{X}_L(\mathcal{C}'_{n-1} - \overline{\mathcal{C}}'_{n-1}) \end{bmatrix}$$

$$- V^{-1}\frac{\gamma\mu}{c}\begin{bmatrix} \mathcal{D}\mathcal{X}_L(\nabla\mathcal{J}(\mathcal{W}'_{n-1}) - \nabla\mathcal{J}(\overline{\mathcal{W}}'_{n-1})) \\ \mathcal{B}^{-1}\mathcal{D}\mathcal{X}_L(\nabla\mathcal{J}(\overline{\mathcal{W}}'_{n-1}) - \nabla\mathcal{J}(\overline{\mathcal{W}}'_n)) \end{bmatrix}. \qquad\text{(S43)}$$

---

[4]To ensure that the diagonal elements of $P$ have an amplitude less than 1, the condition $\lambda_i > -\frac{1}{3}$ must hold. This condition relaxes the positive definiteness requirement of the combination matrix in Assumption 1 and imposes the constraint $A > -\frac{1}{3}I_K$ for the EDSG-RMS.

### B.2 PROOF OF THEOREM 2

Using the definition in (S39), we have

$$\|\boldsymbol{\mathcal{W}}_n' - \overline{\boldsymbol{\mathcal{W}}}_n'\|^2 \leq c^2 \|\mathcal{X}_R\|^2 \|V_U\|^2 \|\boldsymbol{h}_n\|^2. \tag{S44}$$

Define the error vector $\overline{\boldsymbol{w}}_n^{e'} = \overline{\boldsymbol{w}}_n' - w^*$. Using (S44) and following a process similar to that in (S14)-(S18), from (S40) we get

$$\mathbb{E}\{\|\overline{\boldsymbol{w}}_n^{e'}|\boldsymbol{\mathcal{F}}_{n-1}\|^2\} \leq (1 - \gamma\mu\frac{\nu}{2})\|\overline{\boldsymbol{w}}_{n-1}^{e'}\|^2 - 2\gamma\mu(J(\overline{\boldsymbol{w}}_{n-1}') - J(w^*))$$

$$+ \gamma\Big(2\mu\delta + 2\gamma\delta^2\mu^2 + \gamma\frac{2\sigma_\mu^2\delta^2}{K}\Big)\|\mathcal{X}_L\|^2\|\mathcal{X}_R\|^2\|V_U\|^2\|\boldsymbol{h}_{n-1}\|^2$$

$$+ \gamma^2\Big(2\mu^2\|\nabla J(\overline{\boldsymbol{w}}_{n-1}')\|^2 + \frac{4\sigma_\mu^2\|\nabla\mathcal{J}(\mathcal{W}^*)\|^2}{K^2}\Big)$$

$$+ \gamma^2\frac{4\sigma_\mu^2\delta^2}{K}\|\overline{\boldsymbol{w}}_{n-1}^{e'}\|^2 + \gamma^2\frac{\theta_\mu^2\sigma_s^2}{K}. \tag{S45}$$

Under inequality conditions (S19) to (S22), from (S45), we have:

- (Case I: $\nu = 0$)

$$\mathbb{E}\{\|\overline{\boldsymbol{w}}_n^{e'}\|^2\} \leq \mathbb{E}\{\|\overline{\boldsymbol{w}}_{n-1}^{e'}\|^2\} - \gamma\mu\big(\mathbb{E}\{J(\overline{\boldsymbol{w}}_{n-1}')\} - J(w^*)\big)$$

$$+ 3\gamma\mu\delta\|\mathcal{X}_L\|^2\|\mathcal{X}_R\|^2\|V_U\|^2\mathbb{E}\{\|\boldsymbol{h}_{n-1}\|^2\}$$

$$+ \gamma^2\frac{4\sigma_\mu^2\|\nabla\mathcal{J}(\mathcal{W}^*)\|^2}{K^2} + \gamma^2\frac{\theta_\mu^2\sigma_s^2}{K}. \tag{S46}$$

- (Case II: $\nu \neq 0$)

$$\mathbb{E}\{\|\overline{\boldsymbol{w}}_n^{e'}\|^2\} \leq (1 - \frac{\gamma\mu\nu}{4})\mathbb{E}\{\|\overline{\boldsymbol{w}}_{n-1}^{e'}\|^2\} - \gamma\mu\big(\mathbb{E}\{J(\overline{\boldsymbol{w}}_{n-1}')\} - J(w^*)\big)$$

$$+ 3\gamma\mu\delta\|\mathcal{X}_L\|^2\|\mathcal{X}_R\|^2\|V_U\|^2\mathbb{E}\{\|\boldsymbol{h}_{n-1}\|^2\}$$

$$+ \gamma^2\frac{4\sigma_\mu^2\|\nabla\mathcal{J}(\mathcal{W}^*)\|^2}{K^2} + \gamma^2\frac{\theta_\mu^2\sigma_s^2}{K}. \tag{S47}$$

For the case I, from (S46), we have

$$\frac{\gamma\mu}{N}\sum_{n=1}^{N}\big(\mathbb{E}\{J(\overline{\boldsymbol{w}}_{n-1}')\} - J(w^*)\big) \leq \frac{\mathbb{E}\{\|\overline{\boldsymbol{w}}_0^{e'}\|^2\}}{N} + 3\gamma\mu\delta\|\mathcal{X}_L\|^2\|\mathcal{X}_R\|^2\|V_U\|^2\frac{1}{N}\sum_{n=1}^{N}\mathbb{E}\{\|\boldsymbol{h}_{n-1}\|^2\}$$

$$+ \gamma^2\frac{4\sigma_\mu^2\|\nabla\mathcal{J}(\mathcal{W}^*)\|^2}{K^2} + \gamma^2\frac{\theta_\mu^2\sigma_s^2}{K}. \tag{S48}$$

From (S43), with Assumptions 4, 3 and 5, we get

$$\mathbb{E}\{\|\boldsymbol{h}_n\|^2\} \leq \Bigg(\|P\| + \frac{4\gamma^2\delta^2\sigma_\mu^2\|\mathcal{D}\|^2\|\mathcal{X}_L\|^2\|\mathcal{X}_R\|^2\|V_U\|^2\|V^{-1}\|^2}{1 - \|P\|}$$

$$+ \frac{2\tau^2\|\mathcal{B}\|^2\|\mathcal{X}_L\|^2\|\mathcal{X}_R\|^2\|V_U\|^2\|V^{-1}\|^2}{1 - \|P\|}$$

$$+ \frac{2\gamma^2\mu^2\delta^2\|\mathcal{X}_L\|^2\|\mathcal{X}_R\|^2\|V_U\|^2\|V^{-1}\|^2\|\mathcal{D}\|^2}{(1 - \|P\|)}$$

$$+ \frac{4\gamma^4\delta^4\mu^2\|\mathcal{X}_L\|^2\|\mathcal{X}_R\|^2\|V_U\|^2\|V^{-1}\|^2\|\mathcal{D}\|^2\|\mathcal{B}^{-1}\|^2}{(1 - \|P\|)}$$

$$\times\Big(\mu^2 + \frac{\sigma_\mu^2}{K}\Big) + 2\gamma^4\delta^4\mu^2\|V^{-1}\|^2\|\mathcal{X}_L\|^2\|\mathcal{X}_R\|^2\|V_U\|^2$$

$$\times \|\mathcal{D}\|^2 \|\mathcal{B}^{-1}\|^2 \Big(\mu^2 + \frac{\sigma_\mu^2}{K}\Big)\Big) \mathbb{E}\{\|\boldsymbol{h}_{n-1}\|^2\}$$

$$+ \Bigg(\frac{8\gamma^4\mu^4\delta^3\|\mathcal{D}\|^2\|\mathcal{B}^{-1}\|^2}{(1-\|P\|)} + \frac{16\gamma^2\delta\sigma_\mu^2\|\mathcal{D}\|^2}{1-\|P\|}$$

$$+ \frac{16\gamma^4\mu^2\sigma_\mu^2\delta^3\|\mathcal{D}\|^2\|\mathcal{B}^{-1}\|^2}{(1-\|P\|)K} + 4\gamma^4\mu^4\delta^3\|\mathcal{B}^{-1}\|^2\|\mathcal{D}\|^2$$

$$+ \frac{8\gamma^4\mu^2\sigma_\mu^2\delta^3\|\mathcal{B}^{-1}\|^2\|\mathcal{D}\|^2}{K}\Bigg)(\mathbb{E}\{J(\overline{\boldsymbol{w}}_{n-1})\} - J(w^*))$$

$$+ \Bigg(\frac{8\gamma^4\mu^2\delta^2\sigma_\mu^2\|\mathcal{B}^{-1}\|^2\|\mathcal{D}\|^2}{(1-\|P\|)K^2} + \frac{8\gamma^2\sigma_\mu^2\|\mathcal{D}\|^2}{(1-\|P\|)K}$$

$$+ \frac{4\gamma^4\sigma_\mu^2\delta^2\mu^2\|\mathcal{B}^{-1}\|^2\|\mathcal{D}\|^2}{K^2}\Bigg)\|\nabla\mathcal{J}(\mathcal{W}^*)\|^2$$

$$+ \Bigg(\frac{2\gamma^4\mu^2\delta^2\theta_\mu^2\|\mathcal{B}^{-1}\|^2\|\mathcal{D}\|^2}{(1-\|P\|)K} + \frac{\gamma^4\mu^2\delta^2\theta_\mu^2\|\mathcal{B}^{-1}\|^2\|\mathcal{D}\|^2}{K} + 2\gamma^2\|\mathcal{D}\|^2\theta_\mu^2\Bigg)\sigma_s^2, \quad \text{(S49)}$$

where $c^2 = K\|\mathcal{X}_L\|^2\|V^{-1}\|^2$ and the following inequalities are used

$$\|\frac{\gamma}{c}\mathcal{D}\mathcal{X}_L(\mu\boldsymbol{I}_{KL} - \boldsymbol{\mathcal{M}}_n)\nabla\mathcal{J}(\mathcal{W}'_{n-1})\|^2 \leq \frac{\gamma^2\sigma_\mu^2}{c^2}\|\mathcal{D}\|^2\|\mathcal{X}_L\|^2\|\nabla\mathcal{J}(\mathcal{W}'_{n-1})\|^2, \quad \text{(S50)}$$

$$\|\nabla\mathcal{J}(\mathcal{W}'_{n-1})\|^2 \leq 2\|\nabla\mathcal{J}(\mathcal{W}'_{n-1}) - \nabla\mathcal{J}(\overline{\mathcal{W}}'_{n-1})\|^2$$

$$+ 4\|\nabla\mathcal{J}(\overline{\mathcal{W}}'_{n-1}) - \nabla\mathcal{J}(\mathcal{W}^*)\|^2 + 4\|\nabla\mathcal{J}(\mathcal{W}^*)\|^2$$

$$\leq 2\delta^2c^2\|\mathcal{X}_R\|^2\|V_U\|^2\|\boldsymbol{h}_{n-1}\|^2 + 8K\delta(J(\overline{\boldsymbol{w}}_{n-1}) - J(w^*)) + 4\|\nabla\mathcal{J}(\mathcal{W}^*)\|^2, \quad \text{(S51)}$$

and

$$\mathbb{E}\{\|\nabla\mathcal{J}(\overline{\mathcal{W}}'_{n-1}) - \nabla\mathcal{J}(\overline{\mathcal{W}}'_n)\|^2|\boldsymbol{\mathcal{F}}'_{n-1}\} \leq 2\gamma^2\mu^2\delta^2K\|\Gamma^\mathsf{T}\nabla\mathcal{J}(\mathcal{W}'_{n-1}) - \nabla J(\overline{\boldsymbol{w}}'_{n-1})\|^2$$

$$+ 2\gamma^2\mu^2\delta^2K\|\nabla J(\overline{\boldsymbol{w}}'_{n-1})\|^2 + \gamma^2\delta^2K\|\Gamma^\mathsf{T}(\mu\boldsymbol{I}_{KL} - \boldsymbol{\mathcal{M}}_n)\nabla\mathcal{J}(\mathcal{W}'_{n-1})\|^2 + \gamma^2\delta^2\theta_\mu^2\sigma_s^2$$

$$\leq 2\gamma^2\Big(\mu^2\delta^4 + \frac{\delta^4\sigma_\mu^2}{K}\Big)c^2\|\mathcal{X}_R\|^2\|V_U\|^2\|\boldsymbol{h}_{n-1}\|^2 + 2\gamma^2\mu^2\delta^2K\|\nabla J(\overline{\boldsymbol{w}}'_{n-1})\|^2$$

$$+ \gamma^2\delta^2\frac{4\sigma_\mu^2\|\nabla\mathcal{J}(\mathcal{W}^*)\|^2}{K} + \gamma^28\sigma_\mu^2\delta^3(J(\overline{\boldsymbol{w}}_{n-1}) - J(w^*)) + \gamma^2\delta^2\theta_\mu^2\sigma_s^2, \quad \text{(S52)}$$

When $\tau$ and $\gamma$ further satisfy

$$|\tau| \leq \frac{1 - \|P\|}{\sqrt{8}\|\mathcal{B}\|\|\mathcal{X}_L\|\|\mathcal{X}_R\|\|V_U\|\|V^{-1}\|}, \quad \text{(S53)}$$

$$\gamma \leq \min\Bigg\{\frac{1 - \|P\|}{8(\sigma_\mu + \mu)\delta\|\mathcal{X}_L\|\|\mathcal{X}_R\|\|V^{-1}\|\|V_U\|},$$

$$\frac{\sqrt{1 - \|P\|}}{3\delta\sqrt{\mu(\mu^2 + \sigma_\mu^2)^{0.5}}\|V^{-1}\|\|\mathcal{B}^{-1}\|\|\mathcal{X}_L\|\|\mathcal{X}_R\|\|V_U\|}, \frac{\sqrt{1 - \|P\|}}{\delta\|\mathcal{B}^{-1}\|\sqrt{12\mu^2 + 2\sigma_\mu^2}}\Bigg\}, \quad \text{(S54)}$$

the inequality recursion (S49) can be simplified to

$$\mathbb{E}\{\|\boldsymbol{h}_n\|^2\} \leq \frac{1 + \|P\|}{2}\mathbb{E}\{\|\boldsymbol{h}_{n-1}\|^2\} + \frac{16\gamma^2\delta(\sigma_\mu^2 + \mu^2)\|\mathcal{D}\|^2}{1 - \|P\|}\big(\mathbb{E}\{J(\overline{\boldsymbol{w}}'_{n-1})\} - J(w^*)\big)$$

$$+ \frac{9\gamma^2\sigma_\mu^2\|\mathcal{D}\|^2}{(1 - \|P\|)K}\|\nabla\mathcal{J}(\mathcal{W}^*)\|^2 + 3\gamma^2\theta_\mu^2\|\mathcal{D}\|^2\sigma_s^2. \quad \text{(S55)}$$

Averaging both sides of (S55) over $N$ iterations gives

$$\frac{1}{N}\sum_{n=1}^{N}\mathbb{E}\left\{\|\boldsymbol{h}_n\|^2\right\} \leq \frac{1}{N}\frac{1+\|P\|}{1-\|P\|}\mathbb{E}\left\{\|\boldsymbol{h}_0\|^2\right\} + \frac{2}{1-\|P\|}\left(\frac{9\gamma^2\sigma_\mu^2\|\mathcal{D}\|^2}{(1-\|P\|)K}\|\nabla\mathcal{J}(\mathcal{W}^*)\|^2\right.$$

$$\left. + 3\gamma^2\theta_\mu^2\|\mathcal{D}\|^2\sigma_s^2\right) + \frac{32\gamma^2\delta(\sigma_\mu^2+\mu^2)\|\mathcal{D}\|^2}{(1-\|P\|)^2}\frac{1}{N}\sum_{n=0}^{N-1}(\mathbb{E}\{J(\overline{\boldsymbol{w}}_n')\} - J(w^*)). \quad \text{(S56)}$$

Substituting (S56) into (S48), we obtain

$$\frac{\gamma\mu}{2}\frac{1}{N}\sum_{n=1}^{N}\left(\mathbb{E}\{J(\overline{\boldsymbol{w}}_{n-1}')\} - J(w^*)\right) \leq \frac{\mathbb{E}\{\|\overline{\boldsymbol{w}}_0^{e\prime}\|^2\}}{N}$$

$$+ 3\gamma\mu\delta\|\mathcal{X}_L\|^2\|\mathcal{X}_R\|^2\|V_U\|^2\left(\frac{1}{N}\frac{2}{1-\|P\|}\mathbb{E}\{\|\boldsymbol{h}_0\|^2\}\right.$$

$$\left. + \frac{18\gamma^2\sigma_\mu^2\|\mathcal{D}\|^2}{(1-\|P\|)^2K}\|\nabla\mathcal{J}(\mathcal{W}^*)\|^2 + \frac{6\gamma^2\theta_\mu^2\|\mathcal{D}\|^2\sigma_s^2}{1-\|P\|}\right)$$

$$+ \gamma^2\frac{4\sigma_\mu^2\|\nabla\mathcal{J}(\mathcal{W}^*)\|^2}{K^2} + \gamma^2\frac{\theta_\mu^2\sigma_s^2}{K}, \quad \text{(S57)}$$

where the following condition is used

$$\gamma \leq \frac{1-\|P\|}{\sqrt{192}\delta\sqrt{\sigma_\mu^2+\mu^2}\|V^{-1}\|\|\mathcal{X}_L\|\|\mathcal{X}_R\|\|V_U\|}. \quad \text{(S58)}$$

By utilizing the definition of $\boldsymbol{h}_0$, reorganize (S57) to obtain the result (19). Furthermore, by integrating the conditions (S53), (S54) and (S58), we get the convergence condition (17).

For the case II (strongly-convex), we have the inequality recursion:

$$\underbrace{\begin{bmatrix}\mathbb{E}\{\|\overline{\boldsymbol{w}}_n^{e\prime}\|^2\}\\\mathbb{E}\{\|\boldsymbol{h}_n\|^2\}\end{bmatrix}}_{t_n'} \preceq \underbrace{\begin{bmatrix}1-\frac{1}{4}\gamma\mu\nu, & 3\gamma\mu\delta\|\mathcal{X}_L\|^2\|\mathcal{X}_R\|^2\|V_U\|^2\\8\gamma^2(\sigma_\mu+\mu)^2\delta^2\frac{\|\mathcal{D}\|^2\|V^{-1}\|^2}{(1-\|P\|)}, & \frac{1+\|P\|}{2}\end{bmatrix}}_{\mathcal{Q}'}$$

$$\times\begin{bmatrix}\mathbb{E}\{\|\overline{\boldsymbol{w}}_{n-1}^{e\prime}\|^2\}\\\mathbb{E}\{\|\boldsymbol{h}_{n-1}\|^2\}\end{bmatrix} + \begin{bmatrix}\gamma^2\frac{4\sigma_\mu^2\|\nabla\mathcal{J}(\mathcal{W}^*)\|^2}{K^2}+\gamma^2\frac{\theta_\mu^2\sigma_s^2}{K}\\\gamma^2\|V^{-1}\|^2\|\mathcal{D}\|^2\left(\frac{9\sigma_\mu^2\|\nabla\mathcal{J}(\mathcal{W}^*)\|^2}{(1-\|P\|)K}+3\theta_\mu^2\sigma_s^2\right)\end{bmatrix}. \quad \text{(S59)}$$

Iterating the inequality, we obtain:

$$t_n' \preceq \mathcal{Q}'^n t_0' + (\boldsymbol{I}_2-\mathcal{Q}')^{-1}\begin{bmatrix}\gamma^2\frac{4\sigma_\mu^2\|\nabla\mathcal{J}(\mathcal{W}^*)\|^2}{K^2}+\gamma^2\frac{\theta_\mu^2\sigma_s^2}{K}\\\gamma^2\|\mathcal{D}\|^2\left(\frac{9\sigma_\mu^2\|\nabla\mathcal{J}(\mathcal{W}^*)\|^2}{(1-\|P\|)K}+3\theta_\mu^2\sigma_s^2\right)\end{bmatrix}$$

$$\preceq \left(1-\frac{\gamma\mu\nu}{8}\right)^n\|t_0'\|_1\cdot\mathbb{1}_2 + (\boldsymbol{I}_2-\mathcal{Q}')^{-1}\begin{bmatrix}\gamma^2\frac{4\sigma_\mu^2\|\nabla\mathcal{J}(\mathcal{W}^*)\|^2}{K^2}+\gamma^2\frac{\theta_\mu^2\sigma_s^2}{K}\\\left(\frac{9\sigma_\mu^2\|\nabla\mathcal{J}(\mathcal{W}^*)\|^2}{(1-\|P\|)K}+3\theta_\mu^2\sigma_s^2\right)\gamma^2\|\mathcal{D}\|^2\end{bmatrix}, \quad \text{(S60)}$$

where the second inequality follows from the condition $\|\mathcal{Q}'\|_1 = \max\left\{1-\frac{1}{4}\gamma\mu\nu + 8\gamma^2(\sigma_\mu^2+\mu^2)\frac{\delta^2\|\mathcal{D}\|^2}{(1-\|P\|)}, 3\gamma\mu\delta\|\mathcal{X}_L\|^2\|\mathcal{X}_R\|^2\|V_U\|^2 + \frac{1+\|P\|}{2}\right\} \leq 1-\frac{1}{8}\gamma\mu\nu$, which holds under the condition:

$$\gamma \leq \min\left\{\frac{\mu\nu(1-\|P\|)}{64(\sigma_\mu^2+\mu^2)\delta^2}, \frac{4(1-\|P\|)}{(24\delta\|\mathcal{X}_L\|^2\|\mathcal{X}_R\|^2\|V_U\|^2+\nu)\mu}\right\}. \quad \text{(S61)}$$

In (S60), we further employ the following inequality:

$$(\boldsymbol{I}_2-\mathcal{Q}')^{-1} \preceq \frac{16}{\gamma\mu\nu(1-\|P\|)}\begin{bmatrix}\frac{1-\|P\|}{2}, & 3\gamma\mu\delta\|\mathcal{X}_L\|^2\|\mathcal{X}_R\|^2\|V_U\|^2\\8\gamma^2(\sigma_\mu^2+\mu^2)\delta^2\frac{\|\mathcal{D}\|^2}{(1-\|P\|)}, & \frac{1}{4}\gamma\mu\nu\end{bmatrix}, \quad \text{(S62)}$$

which necessitates the condition:

$$\gamma \leq \frac{(1 - \|P\|)}{8\sqrt{6}\delta\|\mathcal{X}_L\|\|\mathcal{X}_R\|\|V_U\|\sqrt{\sigma_\mu^2 + \mu^2}}\sqrt{\frac{\nu}{\delta}}. \qquad (S63)$$

By applying this inequality in (S60), the desired result is obtained. By collecting the required conditions on $\gamma$ together with the relations $\|\mathcal{P}\| = \|\mathcal{D}\|$, $\|V\|^2 \leq 4$, and $\|V^{-1}\|^2 \leq \sigma_b^{-1}$, we obtain condition (20).