# OpenReview forum: "Dual Privacy Protection in Decentralized Learning"
_ICLR.cc/2026/Conference — Submitted to ICLR 2026_

### Official Review · Reviewer_BQ2h · 2025-10-27

**Soundness:** 2
**Presentation:** 2
**Contribution:** 2
**Rating:** 2
**Confidence:** 4

**Summary:**

This paper proposes two differential-privacy algorithms, DSG-RMS and Enhanced DSG-RMS, to protect shared model parameters and local gradients in distributed optimization. The authors present convergence results for convex and strongly convex objective functions, respectively. Experimental results on distributed filtering, distributed learning, and target localization tasks demonstrate the effectiveness of the proposed approaches.

However, this paper is poorly written, which makes it difficult to follow. Moreover, the convergence results lack theoretical proofs (no appendix is included—perhaps the authors forgot to attach it), making it impossible to verify the correctness of the results given in Theorem 1 and Theorem 2. Furthermore, the paper provides no analysis of the privacy-protection strength/level of the proposed approaches, either qualitatively or quantitatively.

**Strengths:**

The experimental section is thorough and helps demonstrate the effectiveness of the proposed algorithms.

**Weaknesses:**

1. **Poorly written:** This paper lacks clarity in both motivation and technical presentation. For example, in the second paragraph of the Introduction, the authors simply list several privacy-preserving methods without discussing their limitations or drawbacks, which makes the motivation of the work unclear. Furthermore, Sections 3.1.1 and 3.1.2 only summarize two existing algorithms without sufficient explanation, making the presentation confusing and not reader-friendly, especially for readers unfamiliar with EDSG.

2. **Lack of proofs for convergence results:** The proofs of Theorem 1 and Theorem 2 are omitted in the manuscript, and no appendix containing them is provided after the main text.

3. **Lack of differential-privacy analysis and discussion:** The authors aim to address privacy protection, however, there is no analysis of the achieved privacy level. The only privacy-related information that I can find comes from the algorithmic design itself. Specifically, the matrix $M_{k}$ is used to protect gradient information, which seems more like a heterogeneous stepsize design (similar to NIDS). Likewise, the use of the parameter $\tau$ to protect the transmitted messages seems more like a modification of the coupling matrix. Both designs appear to leverage constructive properties of distributed algorithms to achieve privacy protection. Nevertheless, the paper provides very limited qualitative or quantitative discussion on the strength or level of the achieved privacy protection, which is inappropriate for a work that claims to provide privacy guarantees.

**Questions:**

See the weaknesses above. In addition, I have the following questions:

1. In the Introduction section, the appearance of the parameter $\gamma$ is confusing, as it has not been defined at that point.

2. The paper lacks sufficient comparison with existing literature, which makes its contributions less clear. The authors should provide a more comprehensive review of existing privacy-protection results on distributed optimization (e.g., the PSGT method referred to by the authors), and analyze their limitations rather than merely stating that ``security risks remain." Such a discussion would better highlight the advantages of the proposed approach.

3. The second algorithm, EDSG, appears to be similar to the NIDS algorithm in [r1]. The authors should clarify the differences between the design of NIDS and that of EDSG-RMS.

[r1] Li Z, Shi W, Yan M. A decentralized proximal-gradient method with network independent step-sizes and separated convergence rates. IEEE Transactions on Signal Processing, 2019, 67(17): 4494-4506.

4. In Fig. 1, why does the MSD of the proposed approaches exhibit sudden jumps?

---

> ### Author Response · Authors · 2025-11-28
> **Please see our response.**
>
> Weaknesses:
> 1. Poorly written: This paper lacks clarity in both motivation and technical presentation. For example, in the second paragraph of the Introduction, the authors simply list several privacy-preserving methods without discussing their limitations or drawbacks….
>
> **Answer:** *We have revised the second paragraph of the Introduction to better articulate the motivation underlying our algorithm design (see the highlighted section in Section I). Specifically, our work addresses privacy risks arising during information exchange: namely, the exposure of network-wide estimates to external eavesdroppers and the leakage of local gradients or sensitive data to curious internal agents.*
>
> *Additionally, we have included further clarification regarding the EDSG algorithm. EDSG is a well-established method belonging to the class of primal-dual distributed optimization algorithms. Compared to gradient-tracking approaches, EDSG offers significantly lower communication overhead—each iteration requires only a single round of variable exchange among neighboring agents. This communication efficiency is a primary reason for our focus on the EDSG framework.*
>
> 2. Lack of proofs for convergence results: The proofs of Theorem 1 and Theorem 2 are omitted in the manuscript, and no appendix containing them is provided after the main text.
>
> **Answer:** We have added an appendix to include the theoretical proof.
>
> 3. Lack of differential-privacy analysis and discussion: The authors aim to address privacy protection, however, there is no analysis of the achieved privacy level…..
>
> **Answer:** *In essence, we achieve gradient privacy through multiplicative noise. Specifically, we model the perturbed observation as $Z = (I + V)S$, where $S\in R^n$ is a random vector representing the gradient, $V \in R^{n\times n}$ is a random matrix independent of $S$ with zero mean ($\mathbb{E}[V] = 0$) and i.i.d. Gaussian entries. Under these conditions, an upper bound on the mutual information between $S$ and $Z$ can be derived as: $I(S;Z) \le \frac12 \log \frac{\det(\Sigma_S + \sigma^2 \mathbb{E}[|S|^2] I_n)}{(\sigma^2 \mathbb{E}[|S|^2])^n}$, where $\Sigma_S=\operatorname{Cov}(S)=\mathbb E[(S-\mu)(S-\mu)^T]=\sigma^2I$, $\mu=\mathbb E[S]$, and $\sigma^2$ denotes the variance of the entries of V. Consequently, $\lim_{\sigma^2 \to \infty} I(S;Z) = 0$. This implies that by increasing the variance of the multiplicative noise, the mutual information—and hence the potential for information leakage—can be made arbitrarily small. Regarding this privacy aspect, a new Theorem 3 and remark 5 have been added to the revised version.*
>
> Questions:
> See the weaknesses above. In addition, I have the following questions:
> 1. In the Introduction section, the appearance of the parameter $\gamma$ is confusing, as it has not been defined at that point.
>
> **Answer:** *In the revised version, we have added the definition for this parameter.*
>
> 2. The paper lacks sufficient comparison with existing literature, which makes its contributions less clear. The authors should provide a more comprehensive review of existing privacy-protection results on distributed optimization (e.g., the PSGT method referred to by the authors), and analyze their limitations rather than merely stating that ``security risks remain." Such a discussion would better highlight the advantages of the proposed approach.
>
> **Answer:** *Based on your feedback, in the revised version, we discuss more differential privacy-preserving distributed algorithms in Sections 1 and 2.*
>
> 3. The second algorithm, EDSG, appears to be similar to the NIDS algorithm in [r1]. The authors should clarify the differences between the design of NIDS and that of EDSG-RMS.
> [r1] Li Z, Shi W, Yan M. A decentralized proximal-gradient method with network independent step-sizes and separated convergence rates. IEEE Transactions on Signal Processing, 2019, 67(17): 4494-4506.
> In Fig. 1, why does the MSD of the proposed approaches exhibit sudden jumps?
>
> **Answer:** *The key differences between our work and Ref. [r1] are as follows:*
>
> *1. (Step-size design): Ref. [r1] employs heterogeneous deterministic step-sizes, whereas our method utilizes a random step-size matrix whose entries are stochastic*
>
> *2. Algorithmic setting: Our algorithm operates in a stochastic gradient framework, which accounts for noisy or sampled gradients; in contrast, Ref. [r1] assumes access to exact (deterministic) gradients.*
>
> *3. Privacy mechanism: In our approach, by using large variance of step-size, gradient privacy is preserved even if curious internal agents knows the mean of the neighbor's step-size. By injecting non-zero vectors into the transmitted values, this can effectively prevent external eavesdroppers from inferring the network's estimated values.*
>
> * The sudden jump occurs because we reset the model parameters—to test the algorithm’s ability to reconverge.*

---

### Official Review · Reviewer_nW3Q · 2025-10-28

**Soundness:** 2
**Presentation:** 1
**Contribution:** 2
**Rating:** 2
**Confidence:** 3

**Summary:**

This paper presents a privacy preserving technique for decentralized learning algorithms. A randomized matrix step size is used to protect against model inversion attacks and a non-zero mean perturbation vector is added to enhance the privacy. The paper additionally provides practical applications.

**Strengths:**

Although the idea of perturbing the step size for privacy is not original (as will be stated in the weaknesses section), to the best of my knowledge, this is the first work that attempts to give theoretical guarantees of the convergence. The paper presents convergence bounds of their method for the convex and strongly convex case under standard optimization assumptions.

**Weaknesses:**

- The authors do not provide any supplementary material that is necessary for the proofs.

- Contrary to what the authors claim, DP already accounts for statistical averaging attacks thanks to composition theorems. Hence, the purpose of the work is not clear.

- The idea of perturbing the step size for privacy is not new. The authors do no cite key research paper such as : Enhancing Privacy Preservation in Federated Learning via Learning Rate Perturbation ICCV 2023.

- Although the methods leverage noise to enhance privacy, no connection seems to be made with DP, and no formalization of the privacy guarantees is provided.

- The network model assumption (Assumption 1) is not well explained in the paper. What does it mean for the (undirected) communication graph to be strongly connected ?

- The authors refer to inference attacks without specifying whether they mean membership inference attacks or model inversion attacks.

- No intuition is given to the choice of adding the non-zero vector in the protection step.

**Questions:**

- Why does it help for the matrix to have non-zero non-diagonal elements ?

- Why is the added vector chosen to depend on $w_k(n-1)$ for the first algorithm, and on $w_k(n-2)$ for the second one ?

---

> ### Author Response · Authors · 2025-11-28
>
> The authors do not provide any supplementary material that is necessary for the proofs.
>
> **Answer:** *The full proofs and derivations supporting our theoretical claims have been provided in the Appendix.*
>
> Contrary to what the authors claim, DP already accounts for statistical averaging attacks thanks to composition theorems. Hence, the purpose of the work is not clear. The idea of perturbing the step size for privacy is not new. The authors do no cite key research paper... ICCV 2023.
>
> **Answer:** *There appears to be a potential misunderstanding, which we clarify below. Consider a standard privacy-preserving model of the form: $Z = D + V$, where $V$ denotes additive noise. As the variance of $V$ grows large, the mutual information $I(D;Z)$ indeed tends to zero, suggesting strong privacy protection. However, in distributed learning settings, this approach faces a critical limitation: as the algorithm converges, the transmitted signal $\phi_k(n)$ becomes nearly stationary, i.e., $\phi_k(n) \approx phi_k(n-1)$. In the LDP, the transmitted value $\phi_k(n)$ is perturbed by independent zero-mean noise $v_k(n)$, yielding observations ${y_k(n)=v_k(n) + \phi_k(n)}$. An adversary with access to a long time series of such observations can estimate the steady-state value $\phi_k(\infty)$ by computing the sample mean. The accuracy of this estimator is inversely proportional to the noise variance—smaller variance yields better estimation, but weaker privacy; larger variance improves privacy but degrades utility.
> Crucially, in distributed optimization, injecting noise with excessively large variance is impractical, as it severely compromises the estimation accuracy of the network-wide algorithm. This fundamental trade-off limits the applicability of conventional LDP in such settings. This fundamental trade-off reveals that the use of zero-mean privacy noise may leak network estimates.*
>
> *We thank the reviewer for pointing out the ICCV 2023 paper. While it also uses randomized step-sizes for privacy, our work differs in three key aspects: 1. Problem setting: We study fully decentralized learning, unlike its centralized/federated setup; 2. Step-size design: our random step-sizes are matrix-valued. Since the matrix-step-sizes are stochastic, individual realizations may contain negative entries—enabling richer dynamics and stronger privacy through multiplicative noise. 3. Theoretical analysis: we provide convergence guarantees for both convex and strongly convex objective functions.*
>
> Although the methods leverage noise to enhance privacy, no connection seems to be made with DP, and no formalization of the privacy guarantees is provided.
>
> **Answer:** *In essence, we achieve gradient privacy through multiplicative noise. Specifically, we model the perturbed observation as $Z = (I + V)S$, where $S\in R^n$ is a random vector representing the gradient, $V \in R^{n\times n}$ is a random matrix independent of $S$ with zero mean ($\mathbb{E}[V] = 0$) and i.i.d. Gaussian entries. Under these conditions, an upper bound on the mutual information between $S$ and $Z$ can be derived as: $I(S;Z) \le \frac12 \log \frac{\det(\Sigma_S + \sigma^2 \mathbb{E}[|S|^2] I_n)}{(\sigma^2 \mathbb{E}[|S|^2])^n}$, where $\Sigma_S=\operatorname{Cov}(S)=\mathbb E[(S-\mu)(S-\mu)^T]=\sigma^2I$, $\mu=\mathbb E[S]$, and $\sigma^2$ denotes the variance of the entries of V. Consequently, $\lim_{\sigma^2 \to \infty} I(S;Z) = 0$， meaning privacy can be arbitrarily strengthened by increasing noise variance.*
>
> The network model assumption (Assumption 1) is not well explained ...?
>
> **Answer:** *Based on your suggestion, this assumption has been elaborated. Strong connectedness means that, for any pair of agents, there exists an undirected path with positive edge weights connecting them, and moreover, at least one agent has a positive self-loop (i.e., $a_{kk} > 0$, for some $k$).*
>
> The authors refer to inference attacks without specifying whether they mean membership inference attacks or model inversion attacks.
>
> **Answer:** * This paper primarily considers model inversion attacks. We have already specified these attacks in the paper.*
>
> No intuition is given to the choice of adding the non-zero vector in the protection step.
>
> **Answer:** *Please see the second answer above.*
>
> Questions:
> Why does it help for the matrix to have non-zero non-diagonal elements?
>
> **Answer:** If using a matrix step size with diagonal elements, it scales the gradient for each element. However, using a matrix with non-zero, non-diagonal elements performs a general linear transformation on the gradient vector, which is more effective at preserving the gradient.
>
> Why is the added vector chosen to depend on w_k(n-1)  for the first algorithm, and on w_k(n-2) for the second one?
>
> **Answer:** This is mainly for algorithm analysis purposes. Based on our simulation results, the second algorithm can also use $w_k(n-1)$, but its theoretical analysis becomes extremely complex.

---

### Official Review · Reviewer_jCVo · 2025-10-31

**Soundness:** 2
**Presentation:** 3
**Contribution:** 2
**Rating:** 4
**Confidence:** 4

**Summary:**

This paper investigates the dual privacy leakage risks in decentralized learning systems: not only local data and gradients are vulnerable to exposure, but the shared model parameters themselves may also reveal sensitive information. To address these two types of risks simultaneously, the authors propose a dual-protection framework. Under convex objectives, convergence guarantees are established for both algorithms, and error bounds explicitly accounting for the influence of network topology are derived. Extensive experiments across various tasks—including distributed filtering, distributed learning, and target localization—demonstrate the practical effectiveness and robustness of the proposed methods.

**Strengths:**

1. The paper introduces a novel framework designed to simultaneously protect local gradient information and model parameters. Unlike conventional zero-mean noise injection schemes—which are vulnerable to statistical averaging attacks—the proposed approach combines random matrix step sizes with non-zero protection vectors to enhance robustness.

2. The manuscript is well-organized and clearly motivated. The authors center their design around the vulnerability of zero-mean noise to statistical averaging attacks in decentralized settings and use multiple remarks to explain the intuition and role of key algorithmic components. Overall, the presentation is coherent and easy to follow.

**Weaknesses:**

1. he paper lacks sufficient originality. The authors argue that existing privacy mechanisms relying on zero-mean noise are vulnerable to statistical averaging attacks and thus propose a dual-protection strategy combining non-zero protection vectors and random matrix step-sizes. However, the paper “Masked Diffusion Strategy for Privacy-Preserving Distributed Learning” has already introduced both non-zero-mean protective noise and random matrix step-size mechanisms for privacy preservation. The two works exhibit highly similar structures, core ideas, algorithmic designs, theoretical analyses, and experimental validations. The main differences lie only in the specific formulations of the random matrices and protection noise, while the overall framework and contributions substantially overlap.

2. The privacy analysis is primarily empirical, demonstrating resistance to DLG and statistical averaging attacks through experiments. However, it lacks formal and quantitative privacy guarantees—such as $ (\epsilon, \delta)$-differential privacy proofs or other theoretical formulations—making the claimed privacy protection less rigorous and theoretically unsupported.

3. The manuscript contains several noticeable typographical and notational errors. For instance, the initialization condition for
$ w_k(0)$ is repeated twice in Section 3.1.1, and there is a subscript error in equation (6d). These issues reduce the clarity and technical precision of the paper.

4. The experimental section lacks sufficient baseline comparisons. The paper presents only the performance of the proposed methods without systematic evaluation against existing state-of-the-art privacy-preserving algorithm. As a result, it is difficult to objectively assess the claimed performance and privacy improvements.

**Questions:**

1. Could the authors further clarify the main differences between this work and “Masked Diffusion Strategy for Privacy-Preserving Distributed Learning”? Both papers appear to employ random matrix step-sizes and non-zero protection vectors. Please specify the key innovations—either in algorithmic design, theoretical development, or practical application—that go beyond existing work.

2. Have the authors considered providing a formal privacy guarantee for the proposed framework? If not, could they discuss whether such theoretical bounds are feasible or planned for future work?

3. The experiments currently lack comparisons with existing privacy-preserving baselines (e.g., MPD-SG or LDP-based methods). Would the authors be able to include such baselines—either in the rebuttal or supplementary material—to objectively demonstrate the claimed performance and privacy improvements?

4. How sensitive are the proposed algorithms to the variance of the random matrix step-sizes? Could the authors provide numerical or visual evidence (e.g., plots or ablation results) to illustrate the impact of step-size randomness on convergence stability?

5. There appear to be repeated or inconsistent notations—for instance, the initialization condition for w_k(0) in Section 3.1.1 and the subscript error in equation (6d).

---

> ### Author Response · Authors · 2025-11-28
> **Please see our response.**
>
> 1. The paper lacks sufficient originality. The authors argue that existing privacy mechanisms relying on zero-mean noise are vulnerable to statistical averaging attacks….
>
> **Answer:** *Please allow us to provide the following explanation.*
> *The reviewer noted that the comparative study added random variables to the transmitted data, which substantially reduced estimation accuracy. In contrast, our method directly introduces non-zero adjustments to the transmitted values, improving the overall accuracy of estimation. We also establish general conditions that broaden the possible choices for these non-zero terms. Furthermore, our approach employs a random matrix step-size in a fully general form, whereas the comparative work limits this to a diagonal matrix. In terms of communication efficiency, our algorithm requires only one round of variable transmission per iteration while still achieving unbiased estimation. Finally, our analysis extends to general convex settings, providing a broader theoretical foundation for the proposed methods.*
>
> 2. The privacy analysis is primarily empirical, demonstrating resistance to DLG and statistical averaging attacks …. However, it lacks formal and quantitative privacy guarantees…making the claimed privacy protection less rigorous and theoretically unsupported.
>
> **Answer:** *Yes, this paper validates its privacy analysis through numerical experiments. In our framework, the random step-size acts as multiplicative noise. Therefore, we can consider the observed signal as Z = (I + V)S, where $S \in R^n$ is a random vector representing the true gradient (or sensitive information), and $V \in R^{n\times n}$ is a random matrix independent of $S$, with zero mean $\mathbb{E}[V] = 0$ and i.i.d. Gaussian entries. For simplicity, assume $\mu=\mathbb{E}[S]$ and $\Sigma_S=\operatorname{Cov}(S)=\mathbb{E}[(S-\mu)(S-\mu)^T]= \sigma^2I_n$. We can obtain an upper bound on the mutual information between S and Z: $I(S;Z) \le \frac12 \log \frac{\det(\Sigma_S + \sigma^2 \mathbb{E}[|S|^2] I_n)}{(\sigma^2 \mathbb{E}[|S|^2])^n}$, where where \sigma^2 denotes the variance of the entries of $V$. Consequently, $\lim_{\sigma^2 \to \infty} I(S;Z) = 0$.This implies that by increasing the variance of the multiplicative noise, the mutual information (the potential information leakage) can be made arbitrarily small.*
>
> 3. The manuscript contains several noticeable typographical and notational errors. ….
>
> **Answer:** *These issues have now been corrected.*
>
> 4. The experimental section lacks sufficient baseline comparisons. ….
>
> **Answer:** *We have incorporated two additional baseline methods for comparison: a local differential privacy (LDP) algorithm and the MDSG algorithm. As the results show, our method remains competitive when compared to these newly added benchmarks.*
>
> Questions:
> 1. Could the authors further clarify the main differences between this work and “Masked Diffusion Strategy for Privacy-Preserving Distributed Learning”? Both papers appear to employ random matrix step-sizes and non-zero protection vectors. Please specify the key innovations—either in algorithmic design, theoretical development, or practical application—that go beyond existing work.
>
> **Answer:** *The key differences are as follows. The paper referenced by the reviewer employs additive protection noise injected directly into the transmitted data, which inevitably degrades algorithmic performance. In contrast, our approach does not rely on additive noise; instead, it only leverages the random step-sizes to achieve gradient privacy. Moreover, our theoretical analysis goes beyond that of the referenced work: while they only consider strongly convex settings, we provide convergence guarantees for both convex and strongly convex scenarios, offering a more comprehensive performance characterization.*
>
> 2. Have the authors considered providing a formal privacy guarantee for the proposed framework?
> 3 …4. …
>
> **Answer:** *In the revised version, we have added privacy analysis of network estimates and gradients. It reveals that gradient information leakage can be effectively suppressed by choosing a sufficiently large variance for the random step-sizes. The new comparison object has been added to the simulation, see Figure 1.
>
> The influence of the variance of the random matrix step-sizes on algorithm performance has already been characterized in Theorems 1 and 2: larger variance leads to increased estimation error. However, this degradation can be effectively mitigated by reducing the parameter $\gamma$. Due to time constraints, the numerical results are not yet finalized; however, we are happy to provide them in an appendix upon completion if requested. Repeated and incorrect symbols have been corrected.*

---

### Official Review · Reviewer_Ba3C · 2025-10-31

**Soundness:** 1
**Presentation:** 1
**Contribution:** 1
**Rating:** 2
**Confidence:** 4

**Summary:**

The paper claims to introduce two novel algorithm for decentralised learning with privacy. After an introduction, the paper introduce the algorithm recap mini assumption and divide very complex formula to characterise the convergence and then perform some numerical experiments

**Strengths:**

Unfortunately I can't find how to fill this section as the current presentation is very messy and does not allow to really understand what are the contribution of the paper

**Weaknesses:**

- the paper seems ignorant of all the related walk on the centralised learning in the past years not mentioning void 2006 no all the walk on gossip algorithm in the past year it remains also completely ignorant to past attacks in decentralised setting
- the result are not proved nor analysed
- the clarity of the paper is particularly poor, making hard to properly evaluate the eventual contribution of the paper
- while claiming privacy in the title and at the beginning of the paper there is no result of proof in this direction in the paper

**Questions:**

It seems unlucky that answers to questions can change the perception of the paper in this current stage so I do not list any question there but authors can choose to comment on the weaknesses listed above.

**Details Of Ethics Concerns:**

My concern is about the possible use of LLM to generate the paper. I have no formal proof but some behaviours listed below seems very unlikely to be explained by human behaviour.
- the paper repeatedly cite Sayed 2022 as if the paper was about decentralised learning. However this is a general textbook that is not at all a meaningful reference for the supposed topic of the paper. In particular the volume two is dedicated to inference and does not tackle decentralised learning
- similarly even a very naive author would be unlikely to cite a 2022 reference for the definition of convex function and the page management, dedicating the page 5 to very basic optimization notions, is highly suspicious
- none of the formula, that are quite involved, are properly introduce and there are no appendixes with proofs.
- the matrix page 4 is particularly ugly I believe that LLM did not "see" the poor result.
- all the sentences are a bit strange especially when reading the whole paper. just providing some cherry picking here
"This gradient is a scaled version of the local data, which means that eavesdroppers can potentially extract sensitive
information about the underlying data by observing the gradient" -> if one observes the data the probability of leakage is certain so it is very strange to say can potentially extract
On page 7 the first option is a matrix that is very sparse with only two non-0 diagonals and then the sparse version is "Sparse Randomized Structure: In addition to the diagonal elements, L off-diagonal entries are randomly selected and assigned values drawn from zero-mean random variables"

---

> ### Author Response · Authors · 2025-11-28
> **Please see our response.**
>
> - *We apologize that the original presentation did not make the paper's novelty clear for the reviewer. Below, we restate the key contributions.*
>
> *First, the main goal of this paper is to attempt to preserve the global network estimates (against external eavesdropping) and local training data (against internal curious agent). To achieve this, our algorithms introduce a stochastic step-size matrix and a bias term dependent on the previous the norm of model parameter. In comparison, the second algorithm (EDSG-RMS) is tailored for heterogeneous (non-IID) data settings. Compared with existing privacy-preserving distributed learning methods, our approach achieves a more effective balance between network estimation accuracy and gradient privacy protection, while also mitigating the risks of global network estimate information leakage to external eavesdroppers.*
>
> *Second, the complete proof and supporting derivations are included in the appendix.*
>
> Weaknesses: the paper seems ignorant of all the related walk on….
>
> **Answer:**
> - *This article primarily focuses on privacy-preserving decentralized learning methods. Thus, the second paragraph of the introduction reviews several well-established approaches to privacy protection. We highlight a key limitation of the zero-mean noise injection technique: although it introduces randomness to conceal individual transmitted data, an eavesdropper can still approximate the true network estimate by applying a time-varying moving average to the intercepted data.*
>
> - *Sorry, we're not entirely sure what you mean by "void 2006". Regarding gossip-based algorithms, we note that their core mechanism relies on a gossip matrix, which forms the foundation of many modern decentralized learning frameworks. In this context, the privacy-preserving methods proposed by Zhang et al. (2018) and He et al. (2018)—which we review in our work—can be regarded as variants of gossip-based schemes adapted to enhance data privacy.*
>
> - *In the new version, we have added the privacy analysis. The simulation results shown in Figures 1(c), 2(b), and 3(c) also demonstrate the improved privacy performance of our approach compared to existing methods.*
>
> - *For completeness, the full proofs and derivations supporting our theoretical claims have been provided in the Appendix.*
>
> *We acknowledge that any inaccuracies in this manuscript are unintentional. We affirm that the work presented here is original and has not been submitted elsewhere.*
>
> **Answers to the sentence problems pointed out by the reviewers** *Please allow us to provide the following explanation.*
>
> *(I) We simply cited references that we were relatively familiar with to support the convex hypothesis. We are quoting the first volume of this book, not the second volume; this is a minor error. Page 5 lists some assumptions, which are in preparation for the analysis in this section, so we have placed these assumptions here. Because the author is relatively familiar with the content of the book (Volume 1), it is cited to support the analytical assumptions presented in this paper. Furthermore, regarding the objective function, it can be either convex or non-convex. It would seem strange to place the convex assumption below formula (1), which is why we consistently place the assumption in the analysis section.*
>
> *We don't quite understand what you mean by "the matrix on page 4 is particularly ugly". We guess you might think the matrix $M_k(n)$ is in the wrong form. In this matrix representation, $Z_k(n)$ and $Z^\prime_k(n)$ are two "triangular blocks". They, along with the main diagonal elements, form an $L\times L$ matrix.*
>
> *(II) (Regarding the first sentence): The gradient represents a scaled form of the local data, specifically $\gamma(d_k(n)−x^⊤_k(n)w)x_k(n)$. Although an eavesdropper obtain the term $(d_k(n)−x^⊤_k(n)w)x_k(n)$, the scalar component $(d_k(n)−x^⊤_k(n)w)$ is still unknown. This quantity tends to be relatively large at the beginning of the algorithm’s iterations and gradually decreases as the iterations proceed. If the eavesdropper simply concatenates the observed values to form a training sequence $[(d_k(1)−x^⊤_k(1)w)x_k(1), (d_k(2) − x^⊤_k(1)w)x_k(2), ..., (d_k(N) − x^⊤_k(N)w)x_k(N)]$, the resulting sequence would not accurately reflect the original data (for example, one-dimensional data $x_k(n)$). For this reason, we used the phrase “can potentially extract” to describe the limited ability of the eavesdropper to recover the original data.*
>
> *(Regarding the second sentence): Our intended meaning is that the main diagonal elements of the matrix are random variables with mean $\mu$. Additionally, we randomly select $L$ other positions in the matrix and assign them zero-mean random variables.*
>
> *At last, we also welcome any feedback regarding potential shortcomings and will address them through additional clarification or revision as needed.*

---

### Meta-Review · Area_Chair_xMy7 · 2025-12-10

**Summary:**

Although the poor writing quality has been taken into account, the paper needs a new review cycle to confirm that the changes made indeed meet the requirements for publication at ICLR. I recommend that the authors focus on preparing their submission properly and not rely on the peer‑review process to carry out this kind of work.

From a scientific standpoint, the paper still lacks a precise examination and a clear positioning relative to the state of the art, as well as solid theoretical results to establish privacy guarantees.

**Reviewer Concerns:**

- Concerns addressed
  - partially: the writing quality has been improved, some missing proofs were added, and privacy analysis has been provided
- Still outstanding
  - Related work on decentralized/gossip learning
  - Proofs and privacy analysis was not provided in the first version. Reviewers were not able to complete their reviews. Authors have sent a poor draft and use the reviewing process to improve their paper and maybe finish their proof. This is not fair.

**Reviewer Scores:**

Because the authors hijacked the review process by submitting a manuscript that was still in draft form and used the reviewers’ time to finish their article, I believe the reviewers would not have changed their scores.

---

### Decision · Program_Chairs · 2026-01-26

Reject